# The influence of firn-layer material properties on surface crevasse propagation in glaciers and ice shelves

Theo Clayton[1], Ravindra Duddu[2,3], Tim Hageman[4], and Emilio Martínez-Pañeda[4,1]

[1]Department of Civil and Environmental Engineering, Imperial College London, London SW7 2AZ, UK
[2]Department of Civil and Environmental Engineering, Vanderbilt University, Nashville, TN 37235, USA
[3]Department of Earth and Environmental Sciences, Vanderbilt University, Nashville, TN 37235, USA
[4]Department of Engineering Science, University of Oxford, Oxford OX1 3PJ, UK

**Correspondence:** Emilio Martínez-Pañeda (emilio.martinez-paneda@eng.ox.ac.uk); Ravindra Duddu (ravindra.duddu@vanderbilt.edu)

**Abstract.** Linear elastic fracture mechanics (LEFM) models have been used to estimate crevasse depths in glaciers and to represent iceberg calving in ice sheet models. However, existing LEFM models assume glacier ice to be homogeneous and utilise the mechanical properties of fully consolidated ice. Using depth-invariant properties is not realistic, as the process of compaction from unconsolidated snow to firn to glacial ice is dependent on several environmental factors, typically leading to a
lesser density and Young's modulus in upper surface strata. New analytical solutions for longitudinal stress profiles are derived, using depth-varying properties based on borehole data from the Ronne ice shelf, and used in an LEFM model to determine the maximum penetration depths of an isolated crevasse in grounded glaciers and floating ice shelves. These maximum crevasse depths are compared to those obtained for homogeneous glacial ice, showing the importance of including the effect of the upper unconsolidated firn layers on crevasse propagation. The largest reductions in penetration depth ratio were observed for
shallow grounded glaciers, with variations in Young's modulus being more influential than firn density (a maximum difference in crevasse depth of 46% and 20% respectively); whereas, firn density changes resulted in an increase in penetration depth for thinner floating ice shelves (95%-188% difference in crevasse depth between constant and depth-varying properties). Thus, our study shows that the firn layer can increase the vulnerability of ice shelves to fracture and calving, highlighting the importance of considering depth-dependent firn-layer material properties in LEFM models for estimating crevasse penetration depths and
predicting rift propagation.

## 1 Introduction

The formation of surface and basal crevasses as a consequence of deformation in ice sheets plays an influential role in glacial mass balance (Colgan et al., 2016). Crevasses are predominantly mode I (tensile) fractures that propagate vertically downwards to the depth at which they stabilise (i.e., maximum crevasse depth), depending on the longitudinal stress state normal to
the fracture surface (Enderlin and Bartholomaus, 2020). Fracture propagation can be further aided by the accumulation of meltwater within surface crevasses, the supply and storage of which can be attributed to supraglacial lakes and firn aquifers (Poinar et al., 2017). This can trigger a process called hydrofracture, wherein the meltwater in the crevasse exerts additional

opening stress on the crevasse walls (Weertman, 1973). If the volume of meltwater is sufficiently large, hydrofracture can cause full thickness crevasse propagation and lead to large scale iceberg calving events from ice shelves (Scambos et al., 2009) and

to increases in basal sliding rates (Selmes et al., 2011). Glacial mass loss caused by the calving events represents one of the leading contributors to global sea level rise (Rignot et al., 2013; Frederikse et al., 2020; Siegert et al., 2020).

The first study to examine crevasse propagation in glaciers and ice sheets was conducted by Nye (1957), who proposed an analytical zero-stress model. The so-called Nye zero-stress model assumed that crevasses are dry and that ice has no resistance to fracture; thus, a crevasse will propagate to a depth at which tensile stresses are completely offset by the compressive

overburden pressure due to gravitational self-weight. The zero-stress was later adapted to accommodate the presence of meltwater in crevasses by Benn et al. (2007). Nevertheless, this simplistic model agreed well with observational results for fields of closely-spaced crevasses (Mottram and Benn, 2009), where neighbouring crevasses provide shielding effects of crack tip singularities (Weertman, 1974). However, it underestimates the depths of isolated crevasses, where this shielding effect is not present. Therefore, linear elastic fracture mechanics (LEFM) models were introduced by van der Veen (1998a, b) to model

these isolated surface and basal crevasses.

LEFM models capture the effects of the stress singularity (assuming that the resulting plastic zone around the crack tip is sufficiently small) by evaluating the stress intensity factor at the crack tip. The role of crack size and orientation, ice thickness and applied loading conditions can be captured, through the evaluation of the stress intensity factor (Jiménez and Duddu, 2018). Using the principle of superposition, the LEFM models include the contributions from the tensile normal stress, lithostatic

compressive stress and the meltwater pressure. The LEFM approach has been combined with full-Stokes models to study surface and basal crevasse propagation in the Thwaites glacier, with results agreeing well with NASA's radar penetration depths (Yu et al., 2017). In addition, LEFM has been used to map the vulnerability of Antarctic ice shelf crevasses subject to meltwater-driven hydrofracture, with projections agreeing well with existing fractures being mapped by neural networks (Lai et al., 2020). The LEFM approach has been successfully combined with boundary element methods, capturing the interactions between

basal and surface crevasses and providing estimates for stability (Zarrinderakht et al., 2023) and evolution of crevasse shape (Zarrinderakht et al., 2022). However, While analytical LEFM models are computationally efficient and can be implemented into numerical ice sheet models (Krug et al., 2014), they do not account for the role of creep deformation nor of depth-varying ice material properties on crevasse propagation (Gao et al., 2023).

In recent years, several studies focused on the development of computational approaches for modelling crevasse propagation

in grounded glaciers and floating ice shelves. Notably, continuum damage mechanics (CDM) approaches were developed to describe more complex thermo-hydro-mechanical phenomena, but they are computationally more intensive compared to zero-stress and LEFM approaches. Such approaches can capture the effects of viscous deformation and long-term stress states on crevasses (Duddu and Waisman, 2012; Jiménez et al., 2017; Duddu et al., 2020) and can be readily implemented into Stokes-based ice flow models (Pralong and Funk, 2005; Sun et al., 2017). Recent studies have developed phase-field fracture models

that formulate LEFM crack propagation criteria within the CDM framework, thus unifying the two approaches (Sun et al., 2021; Clayton et al., 2022). Additionally, CDM-based models can be useful in assessing the accuracy of LEFM and zero-stress models and for understanding the conditions under which their predictions are valid (Duddu et al., 2020).

While the above-mentioned LEFM and CDM studies capture a range of mechanical interactions, they assume glacial ice to be a homogeneous material, with values of mechanical properties taken as constants equal to that of fully consolidated ice. In reality, glacial ice forms from the accumulation of snowfall at the upper surface and undergoes compaction as a result of the overburden pressure, the rate of which is dependent on accumulation rates and surrounding temperatures (Veldhuijsen et al., 2022). Moreover, snowflakes are restructured into smaller ice crystals due to wind, which then deform into more stable, compact crystal arrangements (Benn and Evans, 2010); this causes large differences in porosity, density, and strength between these top snow layers, referred to as firn, and deeper glacial ice. Neglecting firn layers leads to an overestimation of mechanical properties in the upper strata (Rist et al., 2002, 1996). This is of particular importance for geological media subject to self-gravitational loading (Paterson, 1994), because the driving stresses are dependent on the mechanical properties in such layered or vertically graded materials. The main hypothesis of this study is that accounting for these depth-dependent material parameters in the LEFM framework would alter the crevasse penetration depths in glaciers and ice shelves.

The aim of this article is to explore the importance of including firn layer effects on surface crevasse propagation. To this end, we determine the maximum crevasse depths in idealized glaciers and ice shelves assuming two different material conditions: fully-consolidated homogeneous ice and vertically graded ice as reported from ice core samples, with depth varying materials properties. We derive the analytical solution for the far field longitudinal stress $\sigma_{xx}$ with depth-dependent properties, which primarily drives the vertical propagation of mode I crevasses. We next systematically investigate the effect of varying each material property in the unconsolidated firn layers through parametric studies as follows: only depth variant density in Section 2.1; only depth variant Young's modulus in Section 2.2); and both depth variant density and Young's modulus in Section 2.3. We then conduct fracture mechanics studies in Section 3 for a grounded glacier using the analytical LEFM models presented in Jiménez and Duddu (2018) and verify its accuracy with the stress-based phase field fracture model of Clayton et al. (2022). Through variations in meltwater depth ratios in water-filled surface crevasses and oceanwater heights in marine-terminating glaciers, the conditions where depth-dependent material properties are influential are explored. In Section 4 these studies are extended to floating ice tongues, Section 5 considers the effects of assuming crevasses to develop rapidly (as linear-elastic) versus slowly developing crevasses (as nonlinear viscous), and we summarize all findings in Section 7.

## 2   Analytical solutions for the longitudinal stress

We consider a grounded glacier, with an idealised rectangular geometry of height $H$ and length $L$. We neglect lateral shear and restrict the domain to a flow line near the terminus with $x$ and $z$ representing the along-flow and vertical coordinates. As the out-of-plane length of the glacier is typically much longer than its length, the plane strain assumption is exploited to reduce the three-dimensional glacier geometry to a two-dimensional (height and length) representation. A visual representation of the geometry and boundary conditions can be found in Fig. 1. Horizontal displacements are restrained at the far left of the domain to prevent rigid body motion and a free slip boundary condition is applied by restraining vertical displacements at the base (represented by rollers). The upper surface is defined as a traction-free surface, representing the atmosphere-ice interface. Load contributions considered are the gravitational self-weight, ocean water pressure at the far right terminus and meltwater pressure

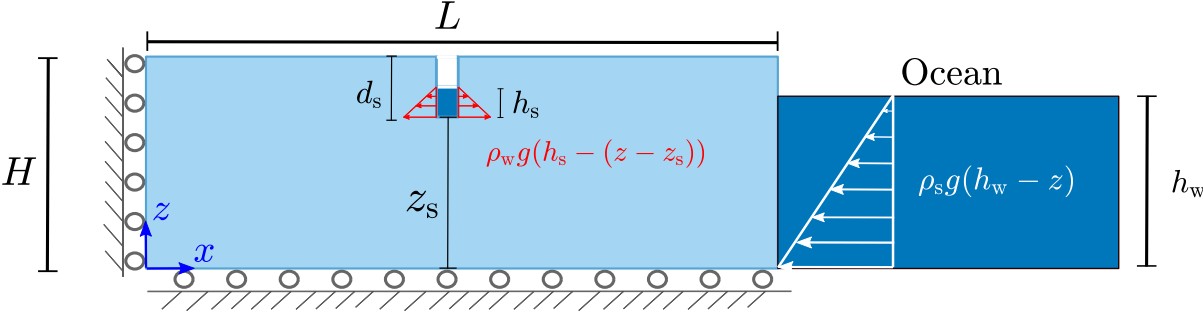

**Figure 1.** Schematic diagram showing the geometry and boundary conditions of a grounded glacier containing a single surface crevasse

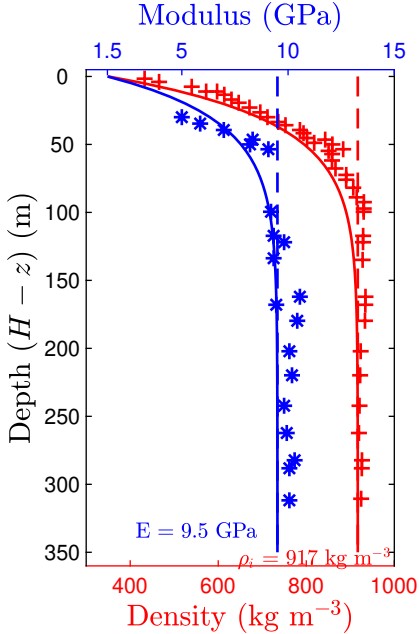

**Figure 2.** Profile of depth-dependent mechanical properties for ice density (red, bottom axis) and Young's modulus (blue, top axis). Data extracted from ice core specimens from the Ronne ice shelf by Rist et al. (2002) are displayed as markers. Isotropic properties are displayed with the dotted lines

in the crevasse. Note that under the small deformation assumption, this representation of the grounded glacier is identical to that of a floating ice shelf Weertman (1957) in the far field region.

The far-field longitudinal stress (based on the long wavelength approximation) within the grounded glacier or ice shelf was derived for the case of steady-state creep with constant (i.e., depth-independent) and homogeneous material properties by

Weertman (1957). Recently, this longitudinal stress was derived for compressible linear elasticity by Sun et al. (2021) as:

$$\sigma_{xx} = \frac{\nu}{(1-\nu)} \rho_i g \left( z - \frac{1}{2} H \right) - \frac{1}{2} \frac{\rho_s g h_w^2}{H} \tag{1}$$

Where $\nu = 0.35$ is the Poisson's ratio, assumed to be equal for firn and ice, $g$ is gravitational acceleration, $\rho_i = 917 \, \text{kg/m}^3$ is the density of fully consolidated glacial ice, $\rho_s = 1020 \, \text{kg/m}^3$ is the seawater density and $h_w$ is the seawater height above the glacier bed. We use the convention that positive $\sigma_{xx}$ corresponds to tensile stresses, creating and propagating crevasses;

whereas negative $\sigma_{xx}$ is compressive, thus stabilising the glacier and preventing the formation of crevasses. This analytical stress will be used throughout this paper to compare between the cases with homogeneous (given by the above equation) and depth-dependent (derived in the remainder of this section) material properties. The Poisson ratio $\nu$ used within our results represents ice as a linear-elastic compressible solid, which is a common assumption for rapidly propagating cracks. If the crevassing process occurs on a time-scale well below the Maxwell time-scale, ranging from hours to days depending on the

strain-rate due to nonlinear viscous nature, the assumption of compressibility would be valid.. If instead the crevassing process occurs slowly, over the span of weeks, the assumption of incompressibility would be valid; so a Poisson ratio of $\nu = 0.5$ will allow for the model derived here to be applicable over longer time-scales. Results for this incompressible case are given in Section 5, whilst results for a depth-dependent Poisson ratio are presented in Appendix E.

As shown in Fig. 2, ice core sample data from the Ronne Ice Shelf, gathered and presented by Rist et al. (2002), indicates

large variations in material properties within the firn and meteoric ice layers forming the upper 150 m of the ice core. The ice core data for density can be fitted using the exponential equation (Paterson, 1994; van der Veen, 1998a):

$$\rho(z) = \rho_i - (\rho_i - \rho_f) e^{-(H-z)/D} \tag{2}$$

where $\rho_f = 350 \, \text{kg/m}^3$ is the density of upper surface unconsolidated firn layers, $H$ is the height of the glacier, $z$ is the vertical coordinate ($z = 0$ at the base of the glacier, $z = H$ at the surface), and $D$ is a constant taken as $D = 32.5 \, \text{m}$ (Rist et al., 2002).

This constant gives an indication of the thickness of the firn layer: at the surface $z = H$ the density of the glacier is equal to that of the unconsolidated firn; at a depth of $H - z = D$ below the surface, the density is in between that of firn and ice, at 75% of the density of ice; and at a depth of $H - z = 2.5D$ below the surface, the density of the glacier is 95% of the density of consolidated ice.

The ice core sample data for elastic modulus from Fig. 2 can be fitted using a similar function as used for the density,

including the depth-dependent Young's modulus as:

$$E(z) = E_i - (E_i - E_f) e^{-(H-z)/D} \tag{3}$$

where $E_i = 9.5 \, \text{GPa}$ is the Young's modulus for solid ice, $E_f = 1.5 \, \text{GPa}$ is the Young's modulus for upper surface unconsolidated firn layers, and $D = 32.5 \, \text{m}$ is a tuned constant. Here we have chosen to use a single constant $D$ to describe both the depth variations in density and Young's modulus. While this is not necessary for the derivation of the analytic solutions,

the interpretation of $D$ as a length scale of the firn layer thickness indicates that Young's modulus is directly proportional to density and inherently related to porosity, similar to porous metallic foams (Ashby et al., 2000).

In the remainder of this section, analytical expressions are derived for the longitudinal stresses driving the creation of crevasses. We first consider two cases wherein solely the density or Young's modulus are depth-varying, so as to isolate and examine the influence of each property on the maximum crevasse penetration depth. For the third and final case, we consider both properties to be depth-varying, which is the real case scenario.

## 2.1 Depth-varying density

We consider only the density is depth-dependent following Eq. (2), while Young's modulus is constant throughout the full thickness (van der Veen, 1998a). The lithostatic compressive stress for this density distribution is:

$$\frac{\partial \sigma_{zz}}{\partial z} = -\rho(z)g \tag{4}$$

with $g$ denoting the gravitational acceleration. By substituting Eq. (2) for the depth-dependent density and integrating over the depth, the vertical stress component is obtained as:

$$\sigma_{zz} = -\rho_i g (H - z) + (\rho_i - \rho_f) Dg \left( 1 - e^{-(H-z)/D} \right) \tag{5}$$

The above relation consists of a linear stress contribution from $\rho_i$ and an exponential term that reduces the net lithostatic stress, when considering the effects of firn density $\rho_f$. This solution simplifies to the homogeneous ice case when considering $\rho_i = \rho_f$.

Exploiting the plane strain assumption $\varepsilon_{yy} = 0$, allows for the out of plane stress $\sigma_{yy}$ to be found in terms of longitudinal stress $\sigma_{xx}$ and lithostatic stress $\sigma_{zz}$:

$$\sigma_{yy} = \nu \left( \sigma_{xx} + \sigma_{zz} \right) \tag{6}$$

Assuming small strains and small rotations, the longitudinal strain can then be written in terms of $\sigma_{xx}$ and $\sigma_{zz}$ by using Hooke's law and Eq. (6):

$$\varepsilon_{xx} = \frac{1}{E} \left[ (1 - \nu^2)\sigma_{xx} - \nu(1 + \nu)\sigma_{zz} \right] \tag{7}$$

Following this, the membrane strain assumption is adopted due to the thickness of the glacier being several orders of magnitude smaller than the length. The longitudinal strain is therefore invariant with depth (Sun et al., 2021):

$$\frac{\partial \varepsilon_{xx}}{\partial z} = 0 \tag{8}$$

Note that the above condition can be derived using Föppl–von Kármán equations describing the large deflections of thin flat plates. Applying this constraint to Eq. (7) allows for the derivative of the horizontal stress to be found:

$$\frac{\partial \sigma_{xx}}{\partial z} = \frac{\nu}{1 - \nu} \frac{\partial \sigma_{zz}}{\partial z} \tag{9}$$

leading to a far-field longitudinal stress of:

$$\sigma_{xx} = \frac{\nu}{1 - \nu} \sigma_{zz} + R_{xx} \tag{10}$$

where $R_{xx}$ is an integration constant that can be interpreted as the depth-invariant tensile resistive stress. Substituting the lithostatic compressive stress $\sigma_{zz}$ from Eq. (5) gives:

$$\sigma_{xx} = \frac{\nu}{1-\nu}\Big(-\rho_i g(H-z)$$
$$+(\rho_i - \rho_f)Dg\big(1 - e^{-(H-z)/D}\big)\Big) + R_{xx} \tag{11}$$

It can be observed that the far-field longitudinal stress is composed of two components: the lithostatic compressive stress component – always negative and thus responsible for crevasse closure; and the resistive tensile stress $R_{xx}$ – responsible for crevasse opening. van der Veen (1998a) conjectured that the inclusion of firn density into the LEFM model would lead to deeper crevasse propagation, because the magnitude of the lithostatic compressive stress would be reduced; however, the reduction in tensile resistive stress was not considered. van der Veen (1998b) stated 'accounting for the lower firn density almost doubles the penetration depth of surface crevasses compared to the constant density model' . To properly account for the influence of firn density, we evaluate the indefinite integration constant $R_{xx}$, appearing in Eq. (10), by considering force equilibrium over the entire thickness in the longitudinal direction as:

$$\int_0^H \sigma_{xx}\, \mathrm{d}z + F_w = 0 \tag{12}$$

where $F_w = \frac{1}{2}\rho_s g h_w^2$ is the hydrostatic force as a result of the ocean water pressure at the glacier front. From this equilibrium between the glaciological longitudinal stress and ocean water pressure at the terminus, we find the resistive tensile stress as:

$$R_{xx} = \frac{\nu}{1-\nu}\frac{\rho_i g H}{2} - \rho_s g \frac{h_w^2}{2H} - \frac{\nu}{1-\nu}(\rho_i - \rho_f)gD$$
$$+ \frac{\nu}{1-\nu}\frac{(\rho_i - \rho_f)gD^2}{H}(1 - e^{-H/D}) \tag{13}$$

Negative values of $R_{xx}$ together with the lithostatic compressive stress prevent crevasse propagation, whereas if $R_{xx}$ exceeds the lithostatic stress only then can crevasses nucleate.

Substituting the value of $R_{xx}$ into the far field longitudinal stress Eq. (11) gives the following analytical solution:

$$\sigma_{xx} = \frac{\nu}{1-\nu}\rho_i g\left(z - \frac{1}{2}H\right) - \frac{1}{2}\frac{\rho_s g h_w^2}{H}$$
$$+ \frac{\nu}{1-\nu}(\rho_i - \rho_f)gD\left(-e^{-(H-z)/D} + \frac{D}{H}(1 - e^{-H/D})\right) \tag{14}$$

This solution simplifies to the one for the far field longitudinal stress in the depth-invariant case, Eq. (1), when considering $\rho_i = \rho_f$. The first term indicates that homogeneous land-terminating glaciers develop crevasses up to half their thickness (based on the zero stress model); whereas the second (negative) term due to the ocean water height reduces the crack driving stress, thereby providing a stabilising effect. The third term includes the influence of depth-dependent density into the stress that is a function of the vertical coordinate $z$; this term is negative near the top surface, thus stabilising the top firn layers, but it tends to

a positive constant in the bottom regions of the glacier, thus increasing the propensity for deeper crevasse propagation below a certain depth. Thus, the inclusion of depth-dependent density can thwart or promote deeper crevasse propagation depending on the glacier and ocean water heights, which is more nuanced than the description by van der Veen (1998a) who neglected any influence of depth-varying density on resistive stress $R_{xx}$.

## 2.2 Depth-varying Young's modulus

We next derive the relation for far-field longitudinal stress, considering a depth variant elastic modulus $E(z)$. The depth-dependent longitudinal strain is given by:

$$\varepsilon_{xx} = \frac{1}{E(z)} \left[ (1 - \nu^2)\sigma_{xx}(z) - \nu(1 + \nu)\sigma_{zz}(z) \right] \tag{15}$$

with $E(z)$ given by Eq. (3). In the depth invariant modulus cases, $E$ simply gets eliminated with Eq. (8), leading to a horizontal stress that is independent of Young's modulus, as given by Eqs. (1) and (14). However, in the depth-dependent modulus case, $E$ does not get eliminated from the far-field longitudinal stress with the membrane strain assumption. However, the longitudinal strain must be depth invariant as required by Eq. (8), so we get:

$$(1 - \nu^2) \frac{E \frac{\partial \sigma_{xx}}{\partial z} - \sigma_{xx} \frac{\partial E}{\partial z}}{E^2} - \nu(1 + \nu) \frac{E \frac{\partial \sigma_{zz}}{\partial z} - \sigma_{zz} \frac{\partial E}{\partial z}}{E^2} = 0 \tag{16}$$

This can then be rearranged to obtain the following expression for the horizontal stress derivative:

$$\frac{\partial \sigma_{xx}}{\partial z} = \frac{\nu}{1 - \nu} \frac{\partial \sigma_{zz}}{\partial z} - \frac{\nu}{1 - \nu} \frac{\sigma_{zz}}{E} \frac{\partial E}{\partial z} + \frac{\sigma_{xx}}{E} \frac{\partial E}{\partial z} \tag{17}$$

This derivation simplifies to the depth invariant case if $\partial E / \partial z = 0$. Solving the above ordinary differential equation yields the following longitudinal stress for constant density:

$$\sigma_{xx} = \frac{\nu}{1 - \nu} \rho_{\mathrm{i}} g \left( z - \frac{(E_{\mathrm{i}} - E_{\mathrm{f}})}{E_{\mathrm{i}}} H e^{-(H-z)/D} \right)$$

$$+ C_1 \left( E_{\mathrm{i}} e^{(H/D)} - (E_{\mathrm{i}} - E_{\mathrm{f}}) e^{(z/D)}, \right) \tag{18}$$

where $C_1$ is an integration constant that can be determined using force equilibrium in the longitudinal direction, defined by Eq. (12), which yields:

$$C_1 = \frac{1}{E_{\mathrm{i}} H e^{H/D} - (E_{\mathrm{i}} - E_{\mathrm{f}}) D \left( e^{H/D} - 1 \right)} \left( \frac{\nu}{1 - \nu} \right.$$

$$\left. \left( \frac{E_{\mathrm{i}} - E_{\mathrm{f}}}{E_{\mathrm{i}}} D \rho_{\mathrm{i}} g H \left( 1 - e^{-H/D} \right) - \frac{\rho_{\mathrm{i}} g H^2}{2} \right) - \frac{\rho_{\mathrm{s}} g h_{\mathrm{w}}^2}{2} \right) \tag{19}$$

Substituting the above into Eq. (18), we obtain the longitudinal stress distribution for the depth-varying Young's modulus and constant density case as:

$$\sigma_{xx} = \frac{\nu}{1 - \nu} \rho_{\mathrm{i}} g \left( z - \frac{1 - E^*}{2} H \right) - \frac{1}{2} (1 + E^*) \frac{\rho_{\mathrm{s}} g h_{\mathrm{w}}^2}{H}$$

$$\text{where } E^* = \frac{(E_{\mathrm{i}} - E_{\mathrm{f}})}{E_{\mathrm{i}}} \frac{(1 - e^{-H/D}) \frac{D}{H} - e^{-(H-z)/D}}{1 - (1 - e^{-H/D}) \frac{(E_{\mathrm{i}} - E_{\mathrm{f}}) D}{E_{\mathrm{i}} H}} \tag{20}$$

As expected, the solution for the variable Young's Modulus case simplifies to the far field longitudinal stress for the depth invariant case, when $E_i = E_f$ ($E^* = 0$). Notably, the variable density only provided an additional term to the homogenous case, but the variable Young's modulus also alters all the terms in the longitudinal stress relation. Including the effects of the firn layer's modulus redistributes the contribution of the horizontal ocean water pressure, where its contribution is lesser near the surface ($E^* < 0$) and greater near the base ($E^* > 0$).

## 2.3 Depth-varying Young's modulus and density

The final stress relationship we derive is for the case with depth varying density and Young's Modulus, using Eq. (2) and (3), respectively. We do not show the full details of the derivation, as it is similar to the previous sections. We use the horizontal stress derivative given in Eq. (17) and substitute the lithostatic compressive stress and derivative from Eqs. (4) and (5). This gives us an ordinary differential equation that is solved using MATLAB's symbolic toolkit to obtain the longitudinal stress $\sigma_{xx}$ as follows:

$$\sigma_{xx} = \frac{\nu}{1-\nu} g \left( -\rho_i (H - z) + (\rho_i - \rho_f) D (1 - e^{-(H-z)/D}) \right)$$
$$+ (E_i e^{H/D} - (E_i - E_f) e^{z/D}) C_2, \tag{21}$$

where $C_2$ is the indefinite integration constant, which can be found using force equilibrium as,

$$C_2 = \frac{1}{E_i H e^{H/D} - (E_i - E_f) D \left( e^{H/D} - 1 \right)} \left( \frac{\nu}{1-\nu} \left( \frac{\rho_i g H^2}{2} \right. \right.$$
$$\left. \left. - (\rho_i - \rho_f) g H D + (\rho_i - \rho_f) g D^2 (1 - e^{-H/D}) \right) - \frac{\rho_s g h_w^2}{2} \right) \tag{22}$$

Substituting $C_2$ back into $\sigma_{xx}$ gives the final expression:

$$\sigma_{xx} = \frac{\nu}{1-\nu} \rho_i g \left( z - \frac{1-E^*}{2} H \right) - \frac{1}{2}(1+E^*) \frac{\rho_s g h_w^2}{H}$$
$$+ \frac{\nu}{1-\nu} (\rho_i - \rho_f) g D \left( \left( 1 - e^{-(H-z)/D} \right) \right.$$
$$\left. + (1 + E^*) \left( -1 + \frac{D}{H}(1 - e^{-H/D}) \right) \right), \tag{23}$$

where the Young's modulus ratio $E^*$ defined in Eq. (20). The above equation reduces to the analytic relation for depth-dependent density in Eq. (14), if the Young's modulus is constant ($E^* = 0$); and to the analytical relation for depth-dependent Young's modulus Eq. (20), if the density is constant ($\rho_i - \rho_f = 0$).

## 2.4 Limitations of analytical LEFM models

There are a few limitations to be noted, regarding the outcomes of the LEFM models used in this study (refer to the Appendices). First, the depth variations of the mechanical properties are assumed based on borehole samples from ice cores in the Ronne ice shelf; therefore may not be fully representative of other Antarctic ice shelves or glaciers elsewhere. For example, temperate

glaciers that are subject to higher rates of melting and refreezing will undergo a faster rate of densification, due to meltwater percolating into pore spaces and refreezing (Wakahama et al., 1976). Thus the process of firn densification is dependent on environmental factors including accumulation rates, overburden pressure, temperature and local strain rates (Oraschewski and Grinsted, 2022). For instance, the Seward Glacier, Yukon, Canada fully consolidates at a depth of $13\,\mathrm{m}$, in contrast to sites at the Greenland ice sheet where transitions from firn to glacial ice occur at depths of $\approx 66\,\mathrm{m}$ (Paterson, 1994). Data from

surrounding borehole samples should therefore be considered when assessing to include the effects of firn layer properties.

  The fracture analysis used in the following sections also assumes that over short timescales, ice behaves as an elastic compressive material ($\nu = 0.35$), with crevasses propagating rapidly in a brittle manner. However, in reality, ice behaves like an incompressible fluid over longer timescales, with viscous deformation being described using Glen's flow law (Glen, 1955). The effects of time-dependent deformation can be included in an *ad hoc* manner by taking the stress from Stokes-based formu-

lations and using LEFM to propagate crevasses in a staggered manner (Yu et al., 2017). Furthermore, the cracking conditions can be more complex than those that can be addressed with a simple analytical model (e.g., multiple crevasse interactions). To account for non-linear behaviour and problems of arbitrary complexity, one can utilize phase field fracture (Clayton et al., 2022), cohesive zone (Gao et al., 2023), or nonlocal creep damage (Huth et al., 2021, 2023) models. However, these nonlinear models are computationally costly and not so easy to implement within numerical ice sheet models, so analytical LEFM

models are desirable.

## 3 Results for grounded glaciers

Fig. 3 shows the analytical solutions of the far field longitudinal stress $\sigma_{xx}$ in a land terminating grounded glacier ($h_{\mathrm{w}} = 0\,\mathrm{m}$) of height $H = 125\,\mathrm{m}$. Each subfigure includes the longitudinal stress versus depth profiles for depth-varying properties (derived in the previous section) and those of the homogeneous case. In addition to the analytical solutions, numerical results from the finite

element solver COMSOL Multiphysics are plotted for verification of our analytic expressions. The analytical solutions derived are identical to the numerical results, confirming the correctness of the presented expressions along with the appropriateness of the membrane strain assumption.

  Using these stress solutions, we predicted crevasse depths based on an analytical LEFM model, as described in the Appendices. For grounded glaciers with free tangential slip at the base, we use the 'double edge crack' weight functions (refer to

Appendix B), as this was shown to yield stress intensity factors that are consistent with those calculated using the displacement correlation method (Jiménez and Duddu, 2018). Specifically, we consider the evolution of an isolated surface crevasse within a grounded glacier of height $H = 125\,\mathrm{m}$, assuming damage to initialise beneath a pre-specified surface crack of depth $d_{\mathrm{s}} = 10\,\mathrm{m}$. To further verify the LEFM results, we conducted phase field fracture (PFF) simulations with the same configuration used for LEFM studies, but with a pre-existing surface notch of depth $d_{\mathrm{s}} = 10\,\mathrm{m}$ and width $b = 2.5\,\mathrm{m}$. The PFF model

simulations capture the nonlinear propagation of the crack driven by the mechanical stress beneath the notch until it reaches a final/stabilised crevasse depth. For a full description of this PFF model, we refer the reader to our recent work (Clayton et al., 2022).

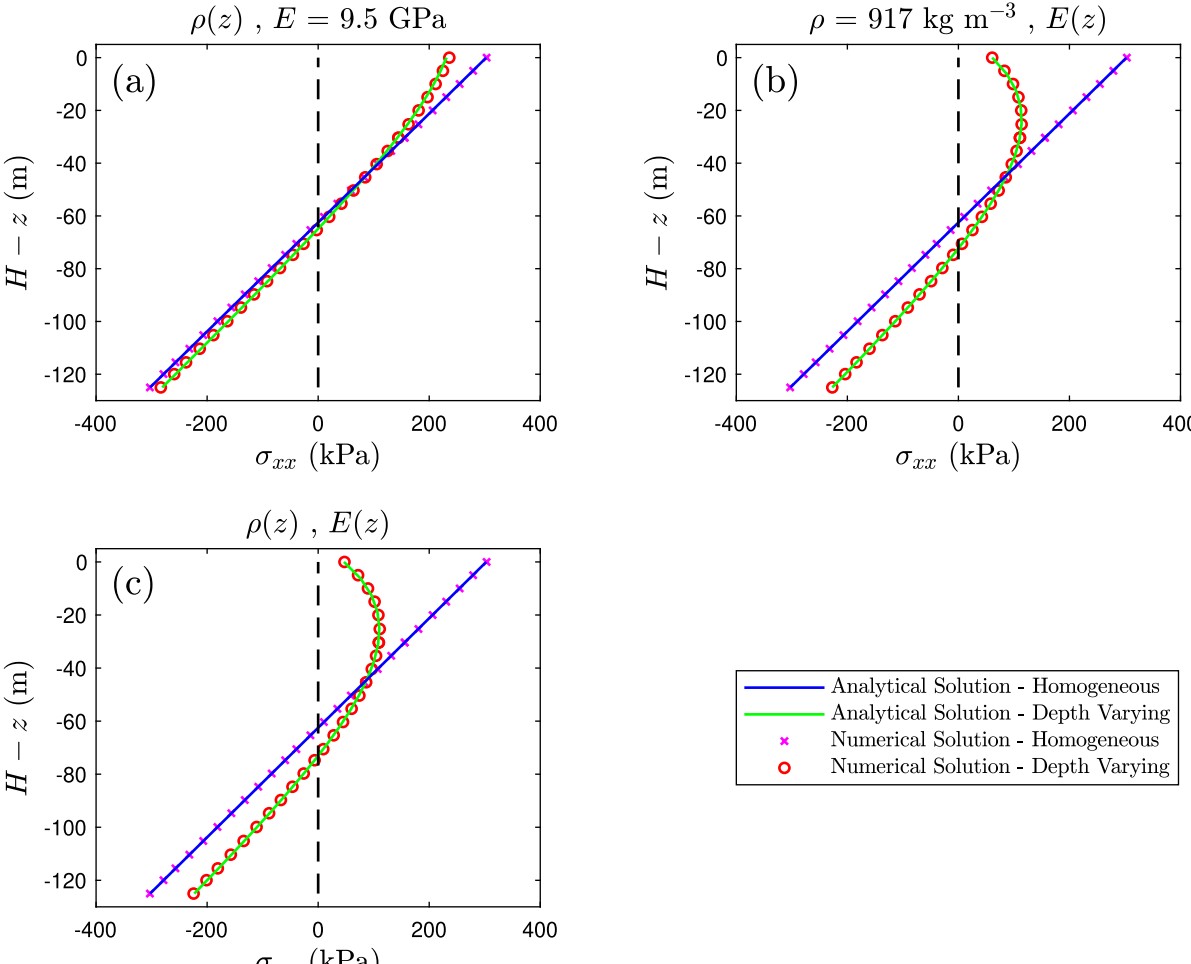

**Figure 3.** Far field longitudinal stress $\sigma_{xx}$ throughout the depth of a land terminating glacier ($h_{\rm w} = 0$), showing the effects of (a) depth-varying density $\rho(z)$, (b) depth-varying Young's modulus $E(z)$, and (c) depth-varying density and Young's modulus $\rho(z), E(z)$. Numerical results are obtained from COMSOL Multiphysics for constant (crosses) and depth-varying properties (circles). The analytical solution for depth-varying properties are given in Eqs. (14), (20) and (23), and represented by the green lines. The vertical dashed line indicates the zero stress level.

The comparison of crevasse depth ratio (i.e., the final crevasse depth normalized with the ice thickness) obtained from LEFM and PFF models is shown in Fig. 4. As the ocean water depth is increased, the glacier is subjected to increased compressive hydrostatic pressure at the glacier terminus, which suppresses crevasse growth. The surface crevasse depth ratio decreases nonlinearly with ocean water depth ratio, which is consistent with previous research works (Bassis and Walker, 2012; Duddu et al., 2013). In each of the depth-varying property cases in Fig. 4, we obtained excellent agreement between LEFM and PFF

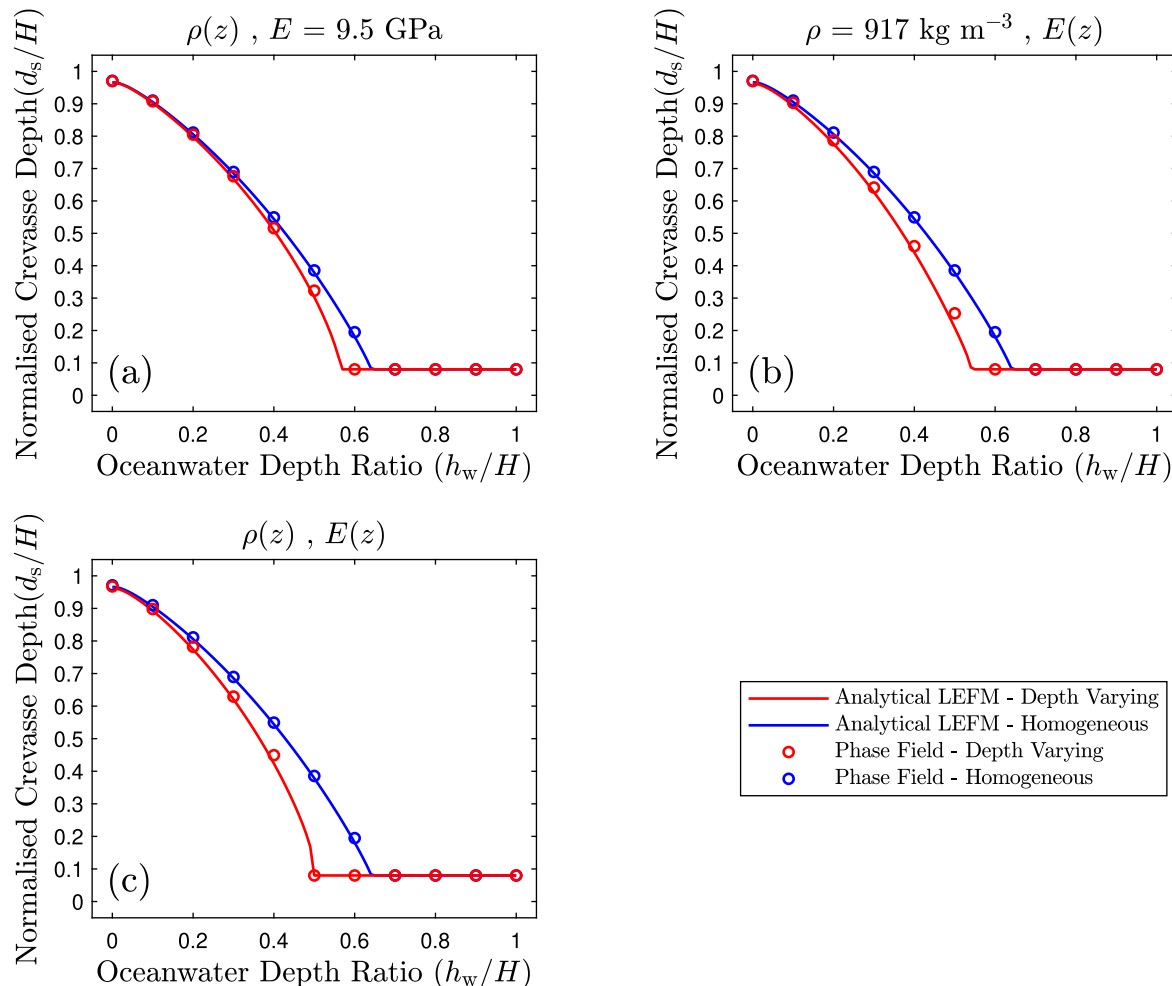

**Figure 4.** Normalised crevasse depth predictions versus oceanwater height ratio for a single isolated dry crevasse in a linear elastic ice sheet, considering homogeneous and depth-dependent mechanical properties

models, which serves as a verification of our LEFM implementation. However, the nuanced differences in crevasse propagation results for each of the cases are discussed in the sections below.

### 3.1 Influence of depth-varying density

If the material properties of ice are assumed to be depth-independent, then the longitudinal stress $\sigma_{xx}$ varies linearly with depth, according to Eq. (1). In the case of a land terminating glacier ($h_w = 0$) with free tangential slip at the base, the $\sigma_{xx}$ profile is symmetric about the centre line ($z = H/2$). This stress is tensile in the regions above the centre line with a maximum

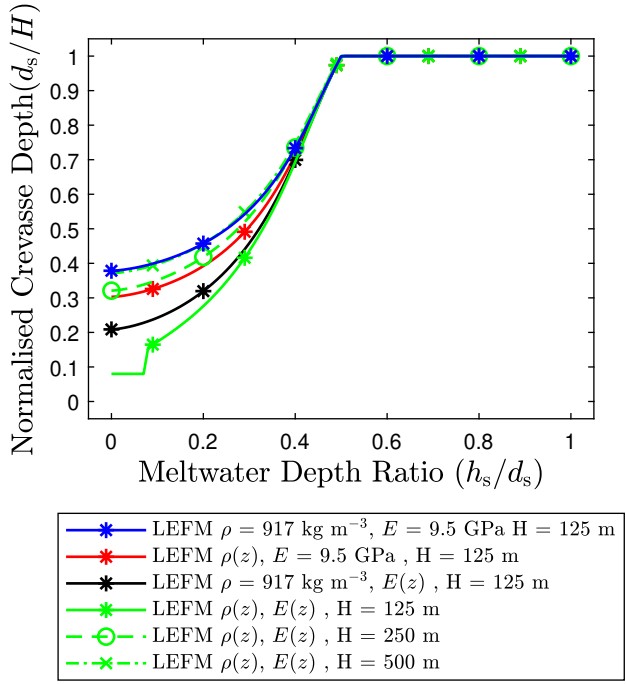

**Figure 5.** Normalised crevasse depth predictions versus meltwater depth ratio for a single isolated crevasse in a linear elastic ice sheet, considering homogeneous and depth-dependent mechanical properties, for an oceanwater height $h_{\mathrm{w}} = 0.5H$

value of $\sigma_{xx} \approx 300\,\mathrm{kPa}$ at the top surface for ice thickness $H = 125$ m (blue line in Fig. 3). If a depth-dependent density is
incorporated (green line in Fig. 3a), the maximum value of $\sigma_{xx}$ is reduced to $\approx 235\,\mathrm{kPa}$, with a non-linear distribution in the upper region. Approximately $50$ m below the top surface, $\sigma_{xx}$ tends towards a linear distribution, with the compressive stress nearer the base slightly less than that compared to the homogeneous case due to the reduced weight of the ice. We next discuss parametric studies to explore the effect of depth varying on crevasse propagation.

      The first parametric study considers a dry (air-filled) crevasse, with different values of ocean water height $h_{\mathrm{w}}$. The normalized
crevasse depths $(d_{\mathrm{s}}/H)$ obtained using LEFM for this case are presented in Fig. 4a. For land terminating glaciers ($h_{\mathrm{w}} = 0$ m), the crevasse propagates to the full thickness of the glacier for both the homogeneous and depth-varying cases, because there is no compressive ocean water pressure to arrest crevasse growth. However, as ocean water height is increased, the stabilised crevasse depth reduces and the inclusion of the depth-varying density comes into effect and further reduces the stabilised crevasse depth. With ocean water heights of $h_{\mathrm{w}} > 0.7H$, the longitudinal stress is compressive enough that the crevasse will
not grow beyond the initial specified depth of 10 m. To verify the accuracy of the LEFM model (solid line) results presented here, we also show the results obtained from the phase field fracture model (markers) in Fig. 4 (Clayton et al., 2022).

In Fig. 5, the relation between the crevasse depth ratio $(d_s/H)$ and the meltwater depth ratio $(h_s/d_s)$ for the thinner glacier $(H = 125\,\text{m})$. The ocean water height is fixed at $h_w = 0.5H$ for both the homogeneous (blue line) and depth-varying density (red line) scenarios. The largest reduction of 20% in the stabilised crevasse depth is observed for a dry crack $(h_s = 0)$ when considering the effect of depth-varying density. The additional tensile stress provided by the presence of meltwater allows the surface crevasse to penetrate deeper into the strata, with full fracture occurring for meltwater depth ratios greater than 0.5 in both scenarios.

As shown in Fig. 3a, the inclusion of the firn layer reduces the longitudinal stress in the upper regions of the glacier, and tends towards the homogeneous stress profile in the consolidated strata. The effect of this stress variation can be understood from Fig. 5. As the meltwater depth ratio is increased, the crevasse penetrates deeper into the glacier and the influence of the firn layer on the stress state disappears; so the crevasse depth for depth-varying case agrees with the homogeneous case for all $h_s/d_s > 0.4$. We found that the normalized crevasse depth ratio is insensitive to the glacier thickness $H$ in the homogeneous case, because the thickness only controls the magnitude of the longitudinal stress but not the depth at which the stress becomes compressive. However, in the depth-varying case, the normalized crevasse depth ratio is sensitive to the glacier thickness, but converges with the homogeneous case for thicker glaciers. For example, the maximum percentage difference in crevasse depth ratio between the depth-varying and homogeneous cases is 20%, 4.5% and 1% for $H = 125\,\text{m}$, $H = 250\,\text{m}$ and $H = 500\,\text{m}$, respectively.

### 3.2 Influence of depth-varying Young's modulus

The influence of a variable Young's modulus on the far field longitudinal stress is shown in Fig. 3b. It can be observed that there is a greater deviation from the homogeneous case, relative to the depth-varying density case. In the upper regions $\sigma_{xx}$ is further reduced to $\approx 60\,\text{kPa}$ and the stress profile is highly non-linear. However, at lower depths where the firn fully consolidates to ice, the stress profile becomes linear. The maximum compressive stress at the glacier base is less due to the reduction in overburden pressure in the upper strata. Notably, the depth at which the stress becomes zero increases from $62.5\,\text{m}$ in the homogeneous case to $72.3\,\text{m}$ in the variable Young's modulus case. However, the stress intensity factor at the crevasse tip decreases due to a reduction in the magnitude of longitudinal stress; thus, the firn layer causes a reduction in crevasse penetration depth.

We now consider the propagation of an isolated dry crevasse for the depth-varying Young's modulus scenario, using the LEFM model and the longitudinal stress relation derived in Eq. (20). The results for the parametric study evaluating the normalized crevasse depths for various ocean water heights $h_w$ are presented in Fig. 4b. Similar to the results in Fig. 4a, the dry crevasse propagates to the same depth in the depth-varying and homogeneous cases for low ocean water heights. Because the crevasse propagates deeper into the fully consolidated ice regions, the properties of firn layer have little impact on crevasse depth. As the ocean water height increases, the compressive stress-resisting crevasse propagation increases, so the crevasse growth is arrested at a shallower depth. The influence of the variable Young's modulus can be observed in these intermediate ocean water heights $(h_w/H = 0.2 - 0.6)$, as the crevasse depth reduces when accounting for the firn layers. The maximum difference in crevasse depth is $\approx 0.2H$, at an ocean water height of $h_w = 0.55H$. For ocean water heights greater than $h_w = 0.55H$, the crevasse does not propagate beyond the initial specified depth of 10 m.

In Fig. 5, we report the normalized crevasse depth ratio versus meltwater depth ratio considering depth-varying Young's modulus (black line). The largest reductions in crevasse depth are observed for the thinner glacier ($H = 125$ m) with a dry crevasse, where $d_s/H$ is reduced from $0.378$ in the homogeneous case to $0.209$ in the depth-varying case. The difference in normalized crevasse depth reduces as the meltwater depth ratio increases because the crevasse penetrates deeper into the fully consolidated strata, thus reducing the influence of firn properties. For thicker glaciers, the difference between the homogeneous and depth-varying Young's modulus cases is smaller, which is attributed to the increase in magnitude of far-field longitudinal stress based on our stress analysis. The maximum percentage difference in crevasse depths for $H = 125$ m is $44.9\%$, $H = 250$ m is $16.5\%$ and $6.0\%$ for $H = 500$ m.

### 3.3 Influence of depth-varying Young's modulus and density

The next set of results entails the propagation of an isolated surface crevasse driven by the longitudinal stress considering both depth-varying Young's modulus and density shown in Eq. (23). The stabilised crevasse depths for a dry crevasse in a grounded glacier of height $H = 125$ m, calculated using LEFM and the phase field method, for various ocean water heights $h_w$ are presented in Fig. 4c. As expected, for lower ocean water heights the crevasse depths are in agreement with the homogeneous case, due to the crevasse penetrating deeper into the compressive regions of the glacier. We observe the largest reductions in crevasse depths for intermediate values of ocean water height ($h_w = 0.2 - 0.4H$); whereas the crevasse does not propagate beyond the initially specified depth for ocean water heights greater than $0.4H$ in this case (red line in Fig. 4c).

The relationship between stabilised crevasse depth ratios and meltwater depth ratios is presented in Fig. 5 for three different glacier thicknesses (green lines). For the thinnest glacier ($H = 125$ m) the longitudinal stress is significantly reduced in the upper regions due to the smaller stiffness and density of the firn layer, which prevents the dry crevasse from propagating beyond the initially specified depth of $0.08$, compared to a crevasse depth ratio of $0.378$ for the homogeneous case. The difference in crevasse depth ratios reduces with increasing meltwater depth ratios, because the crevasse propagates deeper into the consolidated ice strata. Full depth propagation is achieved for meltwater depth ratios $h_s/d_s \geq 0.5$. The percentage difference in penetration depth for the dry crevasse is $18.0\%$ for $H = 250$ m and $6.21\%$ for $H = 500$ m. Overall, we find that the influence of depth-varying firn material properties is lesser in thicker glaciers, however, the effect is more prominent if the variation in both Young's modulus and density with depth is considered.

## 4 Results for floating ice-shelves

The final set of results entails the propagation of surface crevasses in floating ice shelves. We consider an idealised rectangular ice shelf geometry of variable height $H$ and length $L = 5000$ m. Three types of external loads act on the ice shelf: gravitational self-weight causing a body force, and ocean water pressure and meltwater pressure in the crevasse causing surface forces. A Robin-type boundary condition is applied at the base of the ice shelf, because the buoyancy pressure is a function of the vertical displacement $u_z$, as given by $\rho_s g (h_w - u_z)$. The far left terminus is constrained to prevent free body motion, the top surface is considered to be traction-free, and the Neumann boundary condition is applied on the right edge to account for the ocean water

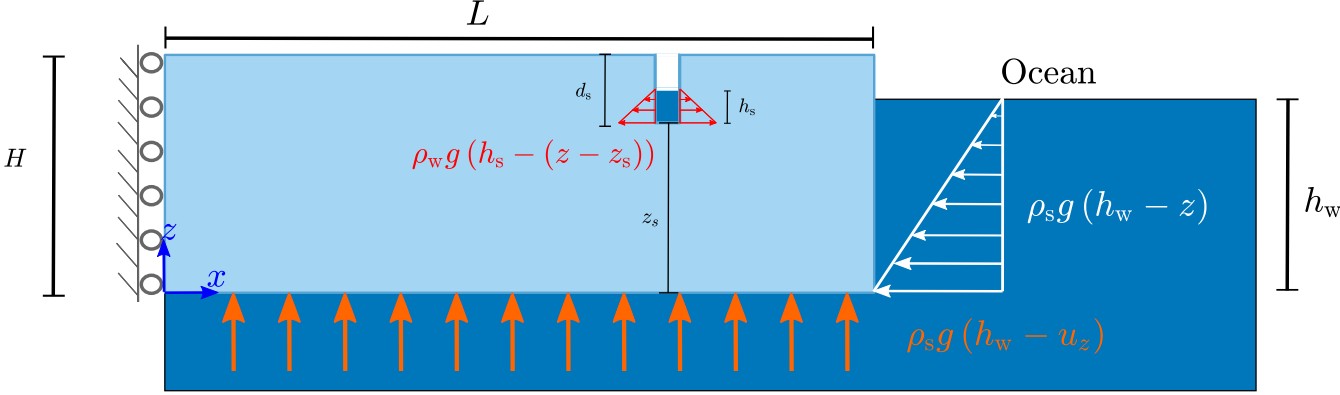

**Figure 6.** Schematic diagram showing the applied boundary conditions of a floating ice shelf containing an isolated surface crevasse

pressure, similarly to the grounded glacier case. A schematic diagram representing the applied boundary conditions is shown in Fig. 6

The floatation heights for the buoyancy pressure are found by assuming local hydrostatic equilibrium. For the homogeneous ice case, this simplifies to the ratio of ice density to ocean water density, that is $h_w/H = \rho_i/\rho_s \approx 0.9$. However, the inclusion of the depth-varying density profile leads to a reduction in the applied gravitational body force, causing a decrease in the floatation height. We evaluate the reduced floatation heights for each ice shelf thickness by integrating the depth-varying density profile over the entire ice shelf thickness and dividing by the thickness and ocean water density. Thus, we obtain that $h_w/H$ is equal

to 0.7560 for $H = 125$ m, 0.8268 for $H = 250$ m, 0.8629 for $H = 500$ m and 0.8809 for $H = 1000$ m. For deeper ice shelves, the material properties of the firn layer become less significant as its thickness is small relative to ice shelf thickness; therefore, the floatation depth tends towards the homogeneous case for thicker ice shelves.

Because of the buoyancy condition at the base and the dependence on ice shelf deflection, the analytical solutions for longitudinal stress derived in Sections 2.1 to 2.3 are not appropriate in the regions closer to the terminus. In Clayton et al.

(2022), we showed that far away from the terminus the longitudinal stress in the ice shelf agreed well with analytical solutions for grounded glaciers. Therefore, we use finite element analysis, to extract the longitudinal stress data as a function of vertical coordinate $z$ at the horizontal position $x = 4750$ m (i.e., 250 m from the ice shelf terminus). The data is fitted to a sixth-order polynomial equation, with coefficients presented in Table C1, which defines the stress function $\sigma_{xx}(z)$. Surface crevasses in the far field region are ignored because the longitudinal stress is significantly compressive, preventing their propagation, regardless

of whether the crevasses were filled with meltwater or not. The propagation of surface crevasses close to the terminus (ice-ocean front) is investigated by estimating the final/stabilized crevasse depth using LEFM. The LEFM model used in Krug et al. (2014) (see Appendix A) is appropriate for floating ice shelves, as this was shown to match numerically calculated stress intensity factors using the displacement correlation method (Jiménez and Duddu, 2018).

Crevasse penetration depths versus meltwater depth ratio $h_s/d_s$ from the LEFM model are presented graphically in Fig. 7.

Penetration depths for the dry crevasse ($h_s/d_s = 0$) and for $h_s/d_s = 0.75$ are also reported in Table 1 and Table 2 respectively,

|  | $\rho = 917 \, \mathrm{kg \, m^{-3}}$, $E = 9.5 \, \mathrm{GPa}$ | $\rho = 917 \, \mathrm{kg \, m^{-3}}$, $E(z)$ | $\rho(z)$, $E = 9.5 \, \mathrm{GPa}$ | $\rho(z)$, $E(z)$ |
|---|---|---|---|---|
| $H = 125$ m | 0.099 | 0.080 (-19.5 %) | 0.250 (151.8 %) | 0.194 (95.0 %) |
| $H = 250$ m | 0.127 | 0.080 (-36.9 %) | 0.203 (59.8%) | 0.174 (37.4%) |
| $H = 500$ m | 0.129 | 0.108 (-16.2%) | 0.167 (28.8%) | 0.156 (20.3%) |
| $H = 1000$ m | 0.121 | 0.114 (-5.8%) | 0.138 (14.0%) | 0.134 (10.7%) |

**Table 1.** Normalised crevasse depths for a dry ($h_\mathrm{s}/d_\mathrm{s} = 0.0$) isolated surface crevasse within a floating ice shelf close to the front (x = 4750 m), calculated using the LEFM method in Krug et al. (2014). Bracketed values represent the difference in crevasse depth between the variational and homogeneous cases normalised by the crevasse depth for homogeneous ice, $(d_\mathrm{s}^{\mathrm{depth\text{-}dep}} - d_\mathrm{s}^{\mathrm{uniform}})/d_\mathrm{s}^{\mathrm{uniform}} \cdot 100\%$

|  | $\rho = 917 \, \mathrm{kg \, m^{-3}}$, $E = 9.5 \, \mathrm{GPa}$ | $\rho = 917 \, \mathrm{kg \, m^{-3}}$, $E(z)$ | $\rho(z)$, $E = 9.5 \, \mathrm{GPa}$ | $\rho(z)$, $E(z)$ |
|---|---|---|---|---|
| $H = 125$ m | 0.317 | 0.294 (-7.2%) | 0.911 (187.1%) | 0.914 (187.9%) |
| $H = 250$ m | 0.362 | 0.359 (-1.0%) | 0.649 (79.2%) | 0.651 (79.8%) |
| $H = 500$ m | 0.370 | 0.370 (0.0%) | 0.512 (38.4%) | 0.513 (38.6%) |
| $H = 1000$ m | 0.375 | 0.374 (-0.3%) | 0.447 (19.2%) | 0.445 (18.7%) |

**Table 2.** Normalised crevasse depths for an isolated surface crevasse with a meltwater depth ratio of $h_\mathrm{s}/d_\mathrm{s} = 0.75$ within a floating ice shelf close to the front (x = 4750 m), calculated using the LEFM method in Krug et al. (2014). Bracketed values represent the difference in crevasse depth between the variational and homogeneous cases normalised by the crevasse depth for homogeneous ice, $(d_\mathrm{s}^{\mathrm{depth\text{-}dep}} - d_\mathrm{s}^{\mathrm{uniform}})/d_\mathrm{s}^{\mathrm{uniform}} \cdot 100\%$

for ease of comparing data points. For the homogeneous ice case, there is minimal influence in normalised crevasse depth from increasing glacier thickness (blue lines); for example the maximum difference in penetration depth for the dry crevasse is $0.022H$ when increasing ice shelf thickness. Surface crevasses with low levels of meltwater only penetrate a few meters below the surface, due to the longitudinal stress profile being predominantly compressive due to the high ocean water pressure. For $h_\mathrm{s}/d_\mathrm{s} < 0.6$, an incremental increase in meltwater does not result in significant crevasse growth. Full fracture propagation is only observed when crevasses are almost fully filled with meltwater ($h_\mathrm{s}/d_\mathrm{s} > 0.9$), where the meltwater pressure is sufficient to overcome the compressive longitudinal stress.

Considering depth-varying Young's modulus leads to a minor reduction in stabilised crevasse depth (black lines), with no growth occurring beyond the initial notch for meltwater depth ratios of $h_\mathrm{s}/d_\mathrm{s} < 0.6$ for $H = 125$ m. Penetration depths match with the homogeneous case for meltwater depth ratios of $h_s/d_s \geq 0.8$ as the crevasse penetrates deeper into the ice strata. The influence of depth-varying Young's modulus reduces with increasing ice shelf thickness. This is the most noticeable for the crevasse depths reported in Table 2, as the percentage difference between the depth-varying modulus and homogeneous cases reduces to 1% for $H = 250$ m.

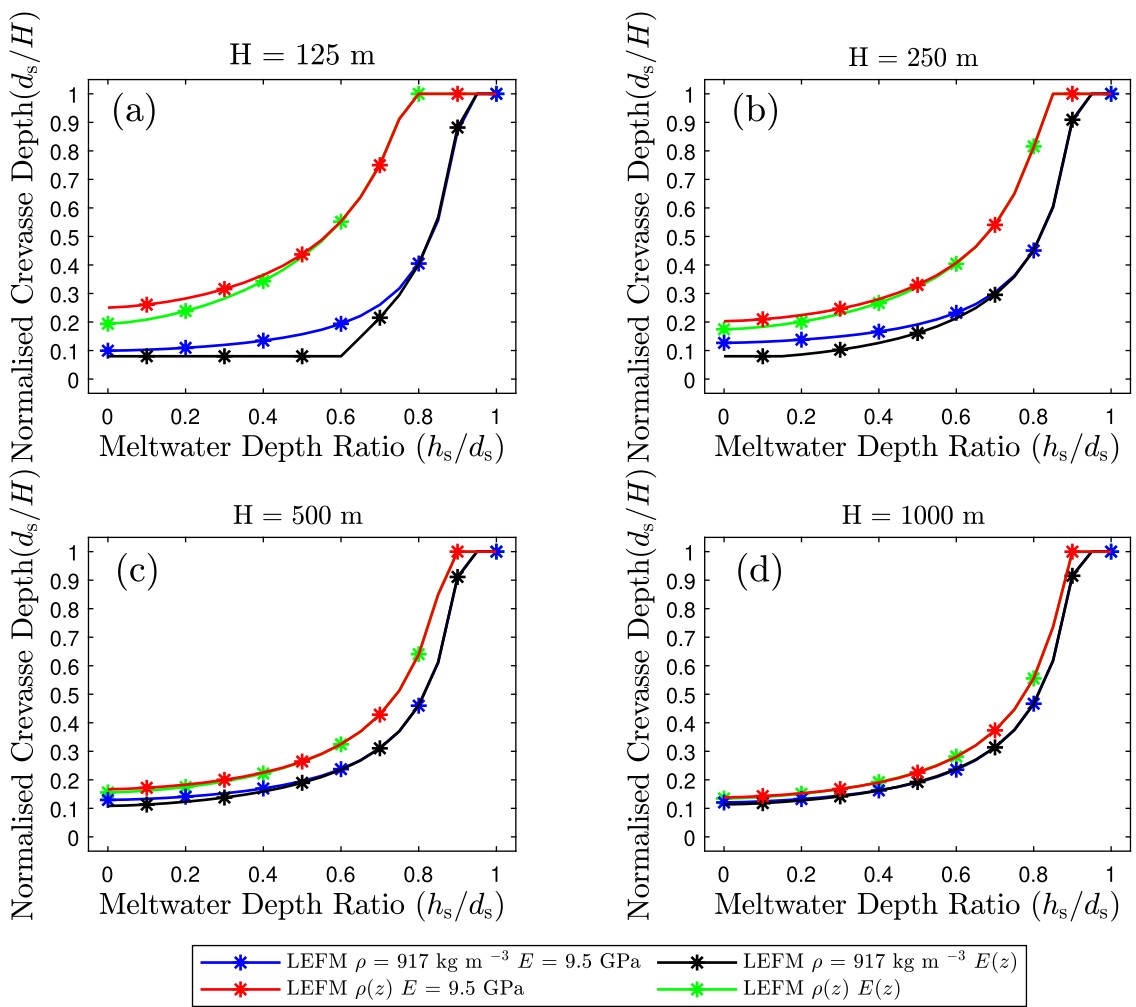

**Figure 7.** Normalised crevasse depth predictions versus meltwater depth ratio for an isolated surface crevasse close to the front in a floating ice shelf.

Considering depth-varying density in the fracture analysis results in surface crevasses propagating to deeper into the ice
strata (red lines), with a dry crevasse propagating to a depth of $0.250H$ compared with $0.099H$ for homogeneous ice within an ice shelf of thickness $H = 125$ m. This is in contrast to the grounded glacier case (where crevasse depth decreased) and can be attributed to the reduction in overburden pressure and the reduced buoyancy height, specific to ice shelves. The inclusion of meltwater within the crevasse results in an increased penetration depth, and full thickness fracture is achieved for meltwater depth ratios of $h_s/d_s \geq 0.8$. The largest differences in penetration depth for the depth-varying density, compared to the homoge-
neous case are observed in Table 2. For thin ice shelves ($H = 125$ m), and a meltwater depth ratio of $h_s/d_s = 0.75$, the crevasse propagates approximately three times deeper when accounting for the depth-varying density. Similarly to the grounded glacier

case, the influence of the depth-varying density is reduced when the ice shelf thickness increases, due to a larger proportion of ice being fully consolidated. However, there are still some differences in penetration depth compared to the homogeneous case for thick ice shelves ($H = 1000$ m), with a percentage difference of $14.0\%$ for the dry crevasse, and $19.2\%$ for $h_s/d_s = 0.75$.

Including the effects of both depth-varying density and depth-varying modulus highlights that density is the more prominent property influencing surface crevasse propagation in ice shelves. It is observed in Fig. 7 that the majority of results for depth-varying density and modulus (green lines) overlap the depth-varying density results (red lines). The exception to this is for dry crevasses in thin ice shelves, where the stabilised penetration depth is $0.194H$ compared to $0.25H$ when considering solely depth-varying density.

## 5 Non-linear Viscous Incompressible Rheology


The above analysis has considered ice to behave as a linear elastic compressible solid, with a Poisson ratio of $\nu = 0.35$. This is a ~~common~~ realistic assumption if crevasse propagation occurs in a rapid brittle manner, such that the cracking occurs on a timescale well below the Maxwell time (typically in the order of hours-days for glacial ice). If the slow development of crevasses is to be considered, with crevasse depths stabilising over a span of weeks, then ice should be considered as

an viscous material. The viscous stress state after a sufficiently long period matches that of an incompressible solid, as the role of the viscous creep on the stresses is to dissipate deviatoric stresses, causing the material to approximate a fluid on these time-scales. This stress state can be achieved by setting the Poisson ratio to $\nu \approx 0.5$ (using $\nu = 0.49$ in our studies to prevent numerical issues). In addition, we conduct finite element simulations for a grounded glacier, including the viscous contributions of ice flow modelled through Glen's flow law, and extract numerical values of the longitudinal stress. These

numerical simulations are used to confirm that the analytic solution for the stress distribution using $\nu \approx 0.5$ matches that of a visco-elastic ice model. To illustrate the influence of ice rheology, we plot the longitudinal stress profile for a land terminating ($h_w = 0$) grounded glacier, considering linear elastic compressibility ($\nu = 0.35$), linear elastic incompressibility ($\nu \approx 0.5$) and a non-linear viscous rheology in Fig. 8.

Firstly, we note that when ice is considered as linear elastic incompressible ($\nu \approx 0.5$), a stress solution is obtained which

matches the steady state creep stress state derived by Weertman (1957) for a depth-independent density, and matches stress profiles obtained through simulations using a visco-elastic rheology. We observe that stresses are more extensional in the upper surface and more compressive at the base when considering incompressibility and that the stress obtained through simulations is independent of ice rheology (Glen's law creep coefficients) once a sufficiently long time has passed. For the homogeneous case, the longitudinal stress varies linearly with depth and is symmetrical about the centre line $z = H/2$. Similarly to the

linear elastic compressible case, the inclusion of depth-dependent density results in a reduction in both the lithostatic stress contribution $\sigma_{zz}$ and the resisitive stress $R_{xx}$, for both material rheologies, a point which was neglected by van der Veen (1998b) who considered $R_{xx}$ to be independent of depth-varying density.

The longitudinal stress profiles presented in Fig. 8 are used to drive crevasse propagation in the linear elastic fracture mechanics study. Values of crevasse penetration depth for an isolated dry crevasse in a grounded glacier, subject to different

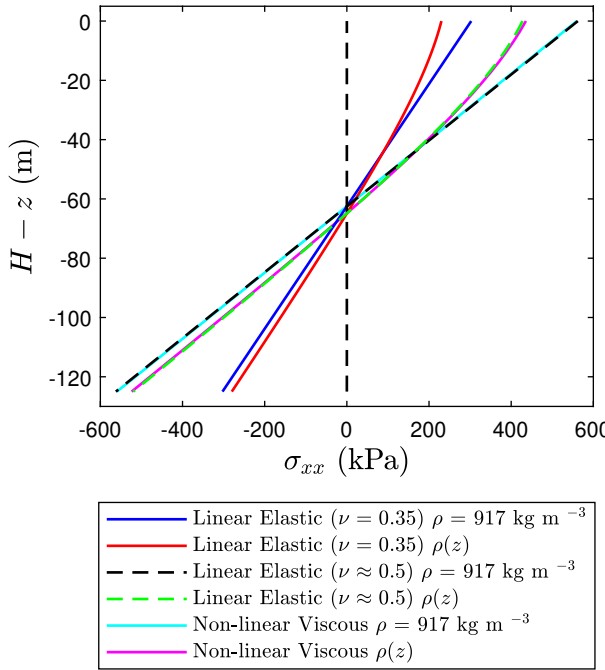

**Figure 8.** Far field longitudinal stress $\sigma_{xx}$ throughout the depth of a land terminating glacier ($h_{\mathrm{w}} = 0$), showing the effects of depth-varying density $\rho(z)$; considering linear elastic compressibility ($\nu = 0.35$), linear elastic incompressibility ($\nu \approx 0.5$) and a non-linear viscous rheology.

values of oceanwater height $h_{\mathrm{w}}$ are presented in Fig. 9. The solid line curves consider incompressible ice, whilst the dashed lines represent compressible ice of Poisson ratio $\nu = 0.35$. Considering ice as an incompressible solid leads to deeper crevasse penetration depths compared to linear elastic compressibility, but these crevasses follow a similar trend as observed for the compressible case: For surface crevasses in glaciers subject to low levels of oceanwater, the penetration depth is unaffected by firn density due to crevasses stabilising in fully consolidated strata. However, as the oceanwater height increases, crevasses be-

come shallower, and as a result, the inclusion of firn density becomes more prevalent. Considering the depth-varying properties of the firn layer in grounded glaciers leads two different regimes of crevasse growth behavior. If the resistive stress $R_{xx}$ is large (e.g. a high oceanwater height), the firn layer promotes crevasse propagation and the crevasse penetrates to a greater depth than in the constant density case. In contrast, if the resistive stress is small, the firn layer has little influence on crevasse propagation. We note that if a Nye-zero stress criterion is used (e.g. for densely spaced cracks) instead of using LEFM to consider a single

isolated crevasse, the firn layer hinders crevasse propagation and the crevasse penetrates to a lesser depth than in the constant density case for a low resistive stress. This can be seen in Fig. 8, where the zero-stress depth in the case of a non-linear viscous model is slightly deeper for the case including the firn density.

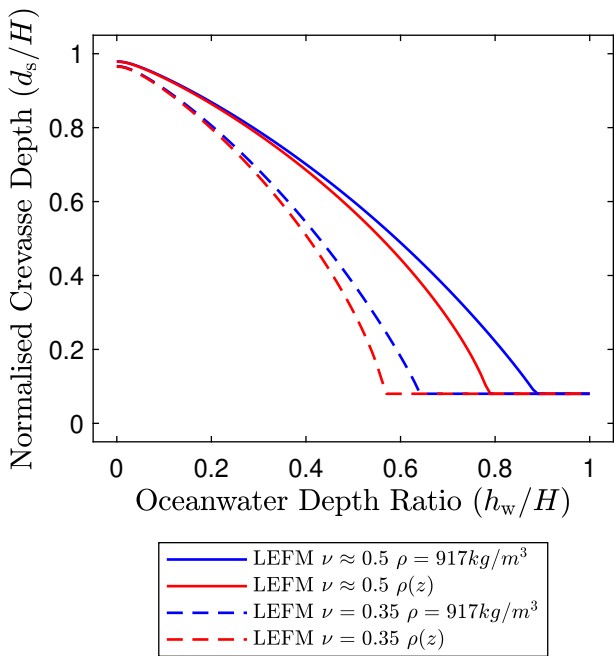

**Figure 9.** Normalised crevasse depth predictions versus oceanwater height ratio for a single isolated dry crevasse in a grounded glacier, considering compressible ($\nu = 0.35$) and incompressible ($\nu \approx 0.5$) ice homogeneous and depth-dependent mechanical properties

Comparing the effects of assuming an incompressible/viscous rheology, the percentage difference in penetration depth when considering depth-dependent density, for a dry crevasse of oceanwater height $h_\mathrm{w} = 0.5H$ reduces to 4%, compared to 20% for linear elastic compressibility. The ocean-water height required to prevent any development of dry crevasses differs, with values of $h_\mathrm{w} = 0.55H$ being sufficient for compressible depth-dependent density cases whereas oceanwater levels of $h_\mathrm{w} = 0.8H$ are required for the incompressible case. Comparing this to the cases in which no density variations are considered still shows a similar trend, with higher oceanwater needed to stabilise crevasses when density variations are not considered.

Finally, we consider water-filled surface crevasses in floating ice shelves of height $H = 125$m and length $L = 5000$m, using a non-linear viscous ice rheology. Similarly to the linear elastic compressible case, we consider surface crevasses at the horizontal position $x = 4750$ m (250 m from the ice shelf terminus) and extract the longitudinal stress profiles from the finite element analysis. We plot the stabilised crevasse depth versus meltwater depth ratio for the non-linear viscous (NLV) rheology in Fig. 10 along with the results for linear elastic (LE) compressibility ($\nu = 0.35$) and incompressibility ($\nu = 0.49$).

When comparing the stabilised crevasse depths close to the front, we note that the penetration depth is independent of ice rheology, which is in contrast to the grounded glacier case. For the homogeneous density, minimal crevasse propagation is observed for meltwater depth ratios below $h_\mathrm{s}/d_\mathrm{s} < 0.6$, with full thickness propagation only occurring when fractures are

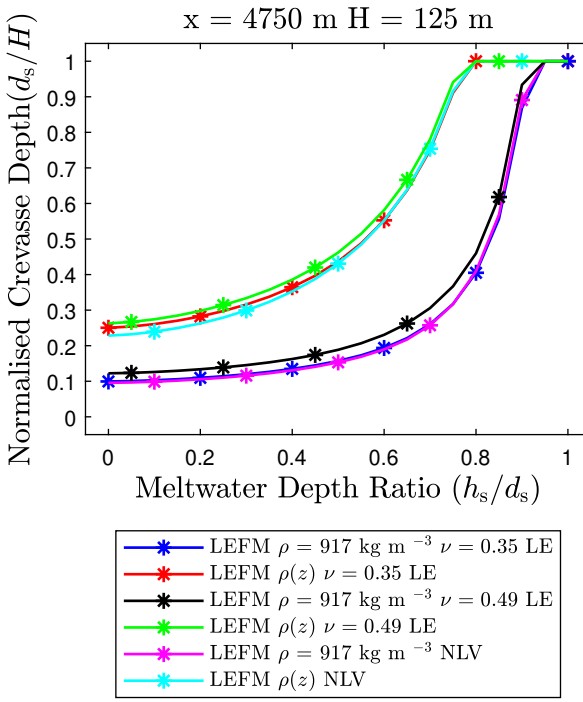

**Figure 10.** Normalised crevasse depth predictions versus meltwater depth ratio for a single isolated surface crevasse located close to the ice shelf front ($x = 4750$ m) considering a linear elastic (LE) and non-linear viscous (NLV) rheology.

close to saturation. The inclusion of the depth-dependent density results in deeper crevasse penetration depths, with minimal differences in penetration depth between the linear elastic cases and non-linear viscous rheology. This likely indicates that for crevasses close to the front, fracture is driven by the flotation height and the bending stresses due to the floating condition.

For depth-dependent density, the reduction in flotation height leads to an increase in tensile stress in the upper surface, due to increases in $R_{xx}$ and increased bending stress. In addition, the lithostatic component of longitudinal stress is reduced, leading to deeper crevasse propagation when including firn density.

We also consider the propagation of an isolated surface crevasse located in the far field region ($x = 2500$ m) of a floating ice shelf, with results presented in Fig. 11. As shown previously, for the linear elastic compressible rheology the stress state is

fully compressive for both the homogeneous and the depth-dependent density case, thus no crevasse propagation is observed regardless of meltwater depth ratio. By contrast, when considering the non-linear viscous rheology of ice, surface crevasses may propagate in the far field region if there is sufficient meltwater pressure present. Large increases in crevasse penetration depth are observed for meltwater depth ratios greater than $h_s/d_s = 0.50$, with full thickness propagation being observed close to crevasse saturation at $h_s/d_s = 0.95$. Similar to crevasses near the front, the inclusion of depth dependent density results in

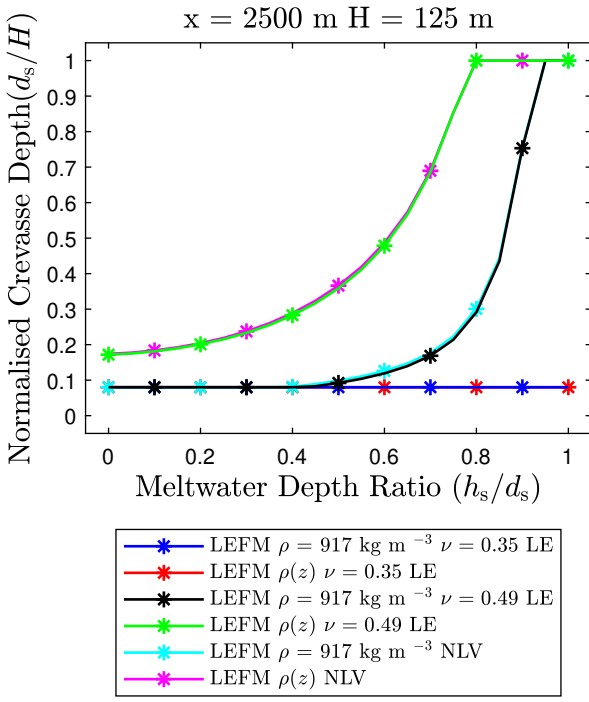

**Figure 11.** Normalised crevasse depth predictions versus meltwater depth ratio for a single isolated surface crevasse located in the far field region ($x = 2500$m) considering a linear elastic (LE) and non-linear viscous (NLV) rheology.

increased crevasse penetration depths compared to the homogeneous density scenario. Thus, similar conclusions can be drawn for both elastic and viscous rheologies.

## 6  Discussion

An important finding of this paper is that the inclusion of the depth-varying mechanical properties of unconsolidated ice strata results in a reduction in both the lithostatic compressive stress and the resistive tensile stress components. Contrary to the 470 conventional understanding (van der Veen, 1998a), we find that accounting for depth-varying density and modulus can lead to an overall reduction in surface crevasse depths in grounded glaciers. This is because, in some scenarios, the reduction in resistive stress can hinder crevasse propagation more than the increase in crevasse propagation resulting from the reduction in lithostatic stress. Thus, our study suggests that firn layers can have a stabilizing effect by curtailing surface crevasse growth in grounded glaciers.

Assuming ice to be an linear elastic compressible material, we find that considering depth-varying Young's modulus has a greater influence on crevasse depths than density in thinner glaciers. For example, considering depth-varying density results in a maximum percentage difference of 20% in the penetration depth of dry crevasses, compared to a maximum percentage

difference of 45% when considering depth-varying Young's modulus. The largest reductions in crevasse depths are observed in thinner glaciers (depths of approximately 100–150 m), where the stabilizing effects of the firn layers seem to be more prominent. Larger meltwater depth ratios are required to propagate surface crevasses in thinner ice shelves; whereas in thicker glaciers, the influence of firn density is lesser (in some cases negligible), so surface crevasses propagate deeper into the fully consolidated strata. Thus, our study reveals that LEFM models assuming homogeneous ice properties are valid for crevasse depth estimation in thicker glaciers with ice thickness $H > 250$ m.

Accounting for depth-varying density in the floating ice shelf case increases the penetration depth of surface crevasses close to the ice-ocean front, with this increase caused by reductions in buoyancy height and lithostatic compressive stresses. The effect of depth-varying density is dominant in thinner ice shelves, but it can still impact surface crevasse propagation in ice shelves as thick as $H = 1000$ m, although to a lesser extend. For instance, the crevasse depth ratio increases to $d_s = 0.91H$ (188% increase compared to homogeneous case) for thin ice shelves ($H = 125$ m); whereas, a 19% increase is observed for 1 km thick ice ($d_s = 0.45H$). Considering depth-varying Young's modulus in the floating ice shelf case slightly reduces surface crevasse depth for low meltwater depths, and the effect becomes less significant in thicker ice shelves. This study suggests as the ice shelves become thinner due to increased basal melting in warmer oceans, the effects of the firn layer can make them more vulnerable by allowing deeper crevasse propagation. However, without the presence of any meltwater, full-depth penetration of surface crevasses in ice shelves may not be possible. Therefore, an important aspect to explore in a future study is the effect of the firn layer on basal crevasse propagation in ice shelves, which likely controls rift formation and iceberg calving.

The analytical (closed-form) and numerical (polynomial-fitted) solutions developed in this paper provide a more realistic description of the longitudinal stress in glaciers and ice shelves. The effect of depth-varying firn/ice properties could be accounted for in the shallow shelf approximation (SSA) and coupled with LEFM models to potentially enable the prediction of rift propagation. Krug et al. (2014) proposed a simple method for estimating crevasse depths in an ice shelf, wherein the longitudinal stress calculated from the full Stokes model was used to evaluate the stress intensity factors based on a LEFM model. A similar approach can be developed to estimate crevasse depths from shallow shelf models or remote sensing data. In Appendix D, we discuss how the 2D stress fields obtained from a shallow ice shelf model can be augmented to include depth-dependent density and Young's modulus. This stress can then be used within an analytical LEFM model to predict crevasse depth and calving.

We acknowledge there are a few limitations to the current study. Firstly, we assume our glacier/ice shelf geometry to be a 2D rectangle with the free slip condition at the base, which is highly idealised and does not consider any contributions from frictional sliding at the base for the grounded case, or buttressing stresses for floating ice shelves (Buck, 2023; Buck and Lai, 2021). Another limitation of the current model is that the rate of firn consolidation is assumed to be uniform with horizontal position and unaffected by glacier thickness $H$. However, firn densification is dependent on environmental factors including accumulation rates, overburden pressure, temperature and local strain rates (van den Broeke, 2008; Amory et al., 2024). Firn densification near terminus regions or in thinner glaciers potentially results in a thinner firn layer and thus a reduced value of the parameter $D$, indicating a shorter length scale for the transition between firn and dense ice properties. However, while the findings presented here are all based on the density profiles from the Ronne ice shelf (Rist et al., 2002), the analytic models

(and provided MATLAB code) allow for an easy way to evaluate the impact for specific firn heights. This makes it possible to estimate the impact of including firn properties on the crevasse depth for specific locations.

One final limitation of our analytic models is related to water-filled crevasses. While we investigate the effects of including the effect of firn on the density and Young's modulus, both these effects are (partially) driven by the porosity of the firn. However, when water-filled crevasses are considered, no model is included to account for water leaking from the crevasse into the surrounding firn. For colder ice-sheets and deeper crevasses, such that the full water contents is surrounded by ice of sub-zero temperatures, this assumption is reasonable: Any water that seeps into the surrounding ice/firn will freeze, creating an

impermeable ice layer surrounding the crevasse which will prevent water from permeating further into the firn (Buzzard et al., 2018; Amory et al., 2024). As these ice layers are typically very thin, they do not alter the mechanical properties of the ice. However, if more temperate glaciers are considered, or conditions where water-filled crevasses do not penetrate to considerable depth, the firn/ice surrounding the crevasse might not be sufficiently cold to cause this ice layer to form. In such circumstances, the presented model will overestimate the crevasse depths obtained, as the saturated firn will reduce the effects of the water

pressure within the crevasse by re-distributing this pressure over a larger region surrounding the crevasse.

## 7   Conclusions

In this paper, we derived analytical equations for the far field longitudinal stress including the effects of surface firn layers, described by depth-varying density and Young's modulus profiles based on field data. These analytic expressions were used to perform fracture propagation studies on isolated air/water-filled surface crevasses in grounded glaciers and ice shelves for the

homogeneous (assuming fully consolidated glacial ice) and depth-varying ice cases. The derived analytical equations for the far field longitudinal stress in grounded glaciers were verified with the stress profiles obtained through finite element analysis. We also used the phase field fracture model to validate the crevasse depths predicted by the LEFM model. The LEFM model results demonstrate the potentially large impact of including the properties of unconsolidated firn layers within the predictions of crevasse depths. For grounded glaciers, depth-varying firn properties inhibit crevasse propagation, requiring lower ocean

water heights and larger meltwater depth ratios for full-depth propagation. In contrast, for floating ice shelves/tongues, the consideration of depth-varying firn properties promoted crevasse propagation, in some cases up to three times as deep. Thus, the firn layer may have a stabilizing effect on grounded glaciers, whereas a destabilizing effect on ice shelves, with regard to fracture and calving. Overall, this study establishes the importance of including the depth-dependence of firn-layer material properties in crevasse models using LEFM. Additionally, we propose a simple scheme to integrate our analytical solutions

with the stresses obtained from the shallow shelf approximation, which can allow us to assess the vulnerability of ice shelves to calving, accounting for firn layer effects.

*Code availability.* The analytic expressions for the stress state are provided in a MATLAB script to plot the stress-depth relations from Fig. 3 for arbitrary input parameters. This additionally performs the LEFM analysis, producing the crevasse depths from Fig. 4. This code


is provided on https://github.com/T-Clayton/LEFM_Firn_Inclusions [NOTE: Github repository currently set to private, will be set to public upon publication of the article].

## Appendix A:  Analytical LEFM model

We evaluate the stress intensity factor (SIF) considering the contributions of normal tensile stress, lithostatic compressive stress and meltwater pressure using an iterative code in MATLAB. An initial crevasse depth $d$ is suggested and the net SIF is found by integrating over the crevasse depth, due to the varying stress field with depth. The crevasse will continue to propagate if $K_{\mathrm{I}}^{\mathrm{net}}$

is larger than the fracture toughness $K_{\mathrm{IC}}$, which for glacial ice is taken as $K_{\mathrm{IC}} = 0.1$ MPa m$^{1/2}$, found from experimental data (Fischer et al., 1995). Since $K_{\mathrm{I}}^{\mathrm{net}}$ is proportional to the net longitudinal stress $\sigma_{\mathrm{net}}$, the SIF will begin to reduce once the crevasse penetrates into the compressive region of the ice. The net longitudinal stress $\sigma_{\mathrm{net}}$ is defined as the sum of the longitudinal stress $\sigma_{xx}$ and the meltwater pressure in the crevasse $p_{\mathrm{w}}$.

$$\sigma_{\mathrm{net}}(z) = \sigma_{xx}(z) + p_{\mathrm{w}}(z), \tag{A1}$$

where

$$p_{\mathrm{w}} = \rho_{\mathrm{w}} g \left\langle h_{\mathrm{s}} - (z - z_{\mathrm{s}}) \right\rangle, \tag{A2}$$

$\rho_{\mathrm{w}}$ is the meltwater density, $z_{\mathrm{s}}$ is the elevation of the crevasse tip above the glacier base and $h_{\mathrm{s}}$ is the meltwater height in the crevasse. The presence of the Macaulay brackets in Eq. (A2) indicates that the pressure is zero above the water surface.

The SIF $K_{\mathrm{I}}^{\mathrm{net}}$ is calculated using the following equation:

$$K_{\mathrm{I}}^{\mathrm{net}} = \int_{0}^{d} M_{\mathrm{D}}\left(\chi, H, d\right) \sigma_{\mathrm{net}}\left(\chi\right) \mathrm{d}\chi, \tag{A3}$$

where $M_{\mathrm{D}}$ is a weight function dependent on the applied boundary conditions and domain geometry, $\chi = H - z$ and $d$ is the trial crevasse depth. The selection of which has been debated in various literature sources (van der Veen, 1998a; Krug et al., 2014).

## Appendix B:  Weight function for grounded glaciers with free tangential slip

For a grounded glacier undergoing free slip, we follow the 'double edge cracks' formulation as this gives good agreement with SIFs calculated using the displacement correlation method with FEM (Jiménez and Duddu, 2018). The weight function is taken from (Tada et al., 1985) and takes the form:

$$M_{\mathrm{D}} = \frac{2}{\sqrt{2H}} \left[ 1 + f_1\left(\frac{\chi}{d}\right) f_2\left(\frac{d}{H}\right) \right] \theta\left(\frac{d}{H}, \frac{\chi}{H}\right), \tag{B1}$$

the functions $f_1$, $f_2$ and $\theta$ are defined as:

$$f_1 = 0.3 \left[ 1 - \left( \frac{\chi}{d} \right)^{\frac{5}{4}} \right], \tag{B2}$$

$$f_2 = \frac{1}{2} \left[ 1 - \sin \left( \frac{\pi d}{2H} \right) \right] \left[ 2 + \sin \left( \frac{\pi d}{2H} \right) \right], \tag{B3}$$

$$\theta = \frac{\sqrt{\tan(\frac{\pi d}{2H})}}{\sqrt{1 - \left[ \frac{\cos(\frac{\pi d}{2H})}{\cos(\frac{\pi \chi}{2H})} \right]^2}}. \tag{B4}$$

## Appendix C: Weight function for floating ice shelves

For the floating ice shelf condition, it was found that the SIFs calculated using the weight function presented in Krug et al. (2014) gave better agreement with SIFs using the displacement correlation method (Jiménez and Duddu, 2018). Therefore, the penetration depths in Section 4 were calculated using the formulation below:

$$K_1^{\text{net}} = \int_0^d \beta \left( z, H, d \right) \sigma_{\text{net}} \left( \chi \right) d\chi \tag{C1}$$

where

$$\beta \left( z, H, d \right) = \frac{2}{\sqrt{2\pi \left( d - z \right)}} \left[ 1 + M_1 \left( 1 - \frac{z}{d} \right)^{0.5} \right.$$
$$\left. + M_2 \left( 1 - \frac{z}{d} \right) + M_3 \left( 1 - \frac{z}{d} \right)^{1.5} \right], \tag{C2}$$

$$\begin{aligned} M_1 = {}& 0.0719768 - 1.513476\lambda - 61.1001\lambda^2 + 1554.95\lambda^3 \\ & - 14583.8\lambda^4 + 71590.7\lambda^5 - 205384\lambda^6 + 356469\lambda^7 \\ & - 368270\lambda^8 + 208233\lambda^9 - 49544\lambda^{10}, \end{aligned} \tag{C3}$$

$$\begin{aligned} M_2 = {}& 0.246984 + 6.47583\lambda + 176.456\lambda^2 - 4058.76\lambda^3 \\ & + 37303.8\lambda^4 - 181755\lambda^5 + 520551\lambda^6 - 904370\lambda^7 \\ & + 936863\lambda^8 - 531940\lambda^9 + 12729\lambda^{10}, \end{aligned} \tag{C4}$$

| | $A$ | $B$ | $C$ | $D$ | $E$ | $F$ | $G$ |
|---|---|---|---|---|---|---|---|
| $\rho = 917$ kg m$^{-3}$, $E = 9.5$ GPa , $H = 125$ m | 0.000 | 0.000 | 0.000 | -0.003 | -0.005 | -1.05 | 0.073 |
| $\rho = 917$ kg m$^{-3}$, $E(z)$, $H = 125$ m | -0.435 | 1.858 | -3.422 | 3.543 | -2.178 | -0.372 | 0.013 |
| $\rho(z)$, $E = 9.5$ GPa, $H = 125$ m | -0.064 | 0.295 | -0.611 | 0.761 | -0.626 | -0.703 | 0.110 |
| $\rho(z)$, $E(z)$ , $H = 125$ m | -0.542 | 2.352 | -4.454 | 4.845 | -3.267 | 0.217 | 0.025 |
| $\rho = 917$ kg m$^{-3}$, $E = 9.5$ GPa , $H = 250$ m | -0.012 | 0.034 | -0.023 | -0.016 | 0.022 | 1.075 | 0.083 |
| $\rho = 917$ kg m$^{-3}$, $E(z)$, $H = 250$ m | -2.688 | 9.862 | -14.657 | 11.290 | -4.749 | -0.074 | 0.017 |
| $\rho(z)$, $E = 9.5$ GPa, $H = 250$ m | -0.453 | 1.711 | -2.652 | 2.207 | -1.083 | -0.725 | 0.100 |
| $\rho(z)$, $E(z)$ , $H = 250$ m | -3.297 | 12.198 | -18.367 | 14.477 | -6.361 | 0.418 | 0.022 |
| $\rho = 917$ kg m$^{-3}$, $E = 9.5$ GPa , $H = 500$ m | 0.189 | -0.728 | 1.113 | -0.819 | 0.254 | -1.086 | 0.082 |
| $\rho = 917$ kg m$^{-3}$, $E(z)$, $H = 500$ m | -6.425 | 21.878 | -29.374 | 19.695 | -6.876 | 0.075 | 0.024 |
| $\rho(z)$, $E = 9.5$ GPa, $H = 500$ m | -0.804 | 2.694 | -3.575 | 2.458 | -1.007 | -0.805 | 0.088 |
| $\rho(z)$, $E(z)$ , $H = 500$ m | -7.599 | 25.940 | -34.968 | 23.630 | -8.396 | 0.406 | 0.027 |
| $\rho = 917$ kg m$^{-3}$, $E = 9.5$ GPa , $H = 1000$ m | 1.246 | -3.946 | 4.596 | -2.235 | 0.216 | -0.945 | 0.067 |
| $\rho = 917$ kg m$^{-3}$, $E(z)$, $H = 1000$ m | -6.490 | 21.657 | -28.390 | 18.540 | -6.324 | -0.028 | 0.030 |
| $\rho(z)$, $E = 9.5$ GPa, $H = 1000$ m | 0.506 | -1.453 | 1.248 | 0.077 | -0.674 | -0.751 | 0.066 |
| $\rho(z)$, $E(z)$ , $H = 1000$ m | -6.858 | 22.931 | -30.191 | 19.901 | -6.927 | 0.129 | 0.031 |

**Table C1.** Coefficients of normalised longitudinal stress in Eq. (C6) for a floating ice shelf at horizontal position ($x = 4750$ m).

$$M_3 = 0.529659 - 22.3235\lambda + 532.074\lambda^2 - 5479.53\lambda^3$$
$$+ 28592.2\lambda^4 - 81388.6\lambda^5 + 128746\lambda^6 - 106246\lambda^7$$
$$+ 35780.7\lambda^8, \tag{C5}$$

and $\lambda = d/H$.

The longitudinal stress in the ice shelf cannot be determined analytically due to the bending stress contribution from the floatation pressure at the base. The stress profiles at the front are obtained numerically (normalised with respect to $\rho_{\mathrm{i}}gH$) and are fitted to a sixth order polynomial equation, taking the general form below:

$$\frac{\sigma_{xx}}{\rho_{\mathrm{i}}gH} = A\left(\frac{\chi}{H}\right)^6 + B\left(\frac{\chi}{H}\right)^5 + C\left(\frac{\chi}{H}\right)^4$$
$$+ D\left(\frac{\chi}{H}\right)^3 + E\left(\frac{\chi}{H}\right)^2 + F\left(\frac{\chi}{H}\right) + G, \tag{C6}$$

where $A, B, C, D, E, F, G$ are non-dimensionalised stress coefficients that are presented in Table C1.

The net SIFs from Eqs. (A3) and (C1) are numerically integrated over the crevasse depth, taking into account the variations in net longitudinal stress with depth. As the crevasse propagates into deeper strata, $K_{\mathrm{I}}^{\mathrm{net}}$ reduces and the crevasse stabilises once $K_{\mathrm{I}}^{\mathrm{net}} = K_{\mathrm{IC}}$.

## Appendix D:  Linking with shallow ice shelf models

Over longer timescales, ice sheet flow is described as an incompressible nonlinear viscous fluid-like model, where the elastic deformations are negligible compared to the viscous deformation. Therefore, the incompressible Stokes equations are used to solve for mass and momentum balance, wherein the volumetric stress or pressure $p$ is constitutively indeterminate and the deviatoric stress tensor $\boldsymbol{\sigma}'$ is defined by a nonlinear viscous constitutive law as:

$$\boldsymbol{\sigma}' = \boldsymbol{\sigma} - p\mathbf{I} = 2\mu(\dot{\varepsilon}_{\mathrm{eq}})\dot{\boldsymbol{\varepsilon}}, \tag{D1}$$

where $\dot{\boldsymbol{\varepsilon}}$ is the strain rate tensor given by the symmetric part of the velocity gradient tensor, $\mathbf{I}$ is the second-order identity tensor, and the viscosity $\mu$ is determined using a Bingham-Norton-Maxwell type relation (Glen, 1955) that accounts for shear thinning behaviour as,

$$\mu = A(T)^{-1/n}\dot{\varepsilon}_{\mathrm{eq}}^{\frac{1-n}{n}}. \tag{D2}$$

In the above equation, $A(T)$ is the temperature-dependent creep coefficient, $n$ is the creep exponent and $\dot{\varepsilon}_{\mathrm{eq}}$ is the second

invariant of the strain rate tensor. Notably, the strain rate tensor can be calculated using surface velocity obtained from remote sensing observations (Chudley et al., 2022).

    While it may not be impossible to simulate ice sheet flow using a full Stokes model, this is currently too computationally expensive. The spatial discretisation in the full Stokes model is dictated by the ice thickness, which is much smaller than the in-plane dimensions of ice sheet/shelf systems, thus leading to an excessively refined mesh for resolving in-plane ice velocity.

The shallow ice shelf approximation (SSA) simplifies the full Stokes model by assuming that vertical shear is zero (MacAyeal, 1989), leading to the following depth-averaged governing equations in the horizontal $x$ and $y$ directions:

$$\frac{\partial}{\partial x}\left(4\overline{\mu}\frac{\partial \overline{u}}{\partial x} + 2\overline{\mu}\frac{\partial \overline{v}}{\partial y}\right) + \frac{\partial}{\partial y}\left(\overline{\mu}\frac{\partial \overline{u}}{\partial y} + \overline{\mu}\frac{\partial \overline{v}}{\partial x}\right) = \rho g H \frac{\partial s}{\partial x},$$
$$\frac{\partial}{\partial y}\left(4\overline{\mu}\frac{\partial \overline{v}}{\partial y} + 2\overline{\mu}\frac{\partial \overline{u}}{\partial x}\right) + \frac{\partial}{\partial x}\left(\overline{\mu}\frac{\partial \overline{v}}{\partial x} + \overline{\mu}\frac{\partial \overline{u}}{\partial y}\right) = \rho g H \frac{\partial s}{\partial y}, \tag{D3}$$

where $\overline{\mu} = \int_0^H \mu \, \mathrm{d}z$ is the depth-integrated viscosity and $s$ is the upper surface elevation. Together with appropriate boundary conditions (e.g. velocity and terminus ocean water pressure), solving this equation provides an excellent approximation of the

flow of ice sheets. Using the velocity components, we can determine the deviatoric stress components according to Eq. (D1). Based on the assumption that the vertical stress $\sigma_{zz}$ is hydrostatic, we can determine the stress components in the plane of the ice shelf as (Greve and Blatter, 2009),

$$\overline{\sigma}_{xx} = 2\overline{\sigma}'_{xx} + \overline{\sigma}'_{yy} - \rho_{\mathrm{i}}g(H-z)$$
$$\overline{\sigma}_{yy} = 2\overline{\sigma}'_{yy} + \overline{\sigma}'_{xx} - \rho_{\mathrm{i}}g(H-z)$$
$$\overline{\sigma}_{xy} = \overline{\sigma}'_{xy}, \tag{D4}$$

where $\overline{\sigma}$ are the height-averaged stresses. Here we propose an approximate way of linking the LEFM model for crevasse depth

estimation with the SSA. Using the in plane stresses in Eq. (D4), we can determine the principal values, which correspond to

the maximum and minimum normal stresses in the plane of the ice shelf. Assuming that crevasses open perpendicular to the maximum principal stress $\overline{\sigma}_{\max}$, we can determine the net force $F_w$ (per unit width out of plane) acting on the boundary as

$$F_w = H\overline{\sigma}_{\max} = H\left(\frac{\overline{\sigma}_{xx} + \overline{\sigma}_{yy}}{2} + \frac{1}{2}\sqrt{(\overline{\sigma}_{xx} - \overline{\sigma}_{yy})^2 + 4\overline{\sigma}_{xy}^2}\right). \tag{D5}$$

The force $F_w$ if compressive would have a similar effect as the ocean water pressure term in Eq. (23), so it can be used to
replace the term $\frac{1}{2}\rho_s g h_w^2$. Thus, the SSA stress solution can be augmented to include the effect of depth varying density and modulus as,

$$\begin{aligned}
\sigma_{xx} &= \frac{\nu}{1-\nu}\rho_i g\left(z - \frac{1-E^*}{2}H\right) - (1+E^*)\frac{F_w}{H} \\
&+ \frac{\nu}{1-\nu}(\rho_i - \rho_f)gD\Big((1 - e^{-(H-z)/D}) \\
&+ (1+E^*)\big(-1 + \frac{D}{H}(1 - e^{-H/D})\big)\Big).
\end{aligned} \tag{D6}$$

The above stress can be used as the net stress to determine the stress intensity factor and estimate the crevasse depth. We note that depending on the discretisation used for the numerical solution of the SSA, some of the details of the scheme would need
modifications. In future work, we will establish the viability of this scheme to couple the LEFM model with the shallow shelf and other depth integrated approximations.

## Appendix E: Influence of Depth-variable Poisson ratio $\nu$

For the crevasse propagation studies previously presented, a depth invariant Poisson ratio of $\nu = 0.35$ was assumed. However, it has been suggested that Poisson ratio also exhibits a linear dependency on ice density and therefore leads to a depth-dependent
profile (Smith, 1965). Furthermore, Schlegel et al. (2019) and King and Jarvis (2007) provides a depth-dependent Poisson ratio profile based on seismic velocity measurements on ice cores. To study the effect of this depth-dependent Poisson ratio, a linear elastic fracture mechanics study is performed. We assume an exponential distribution of Poisson's ratio with depth, similar to the density and Young's modulus distributions.

$$\nu(z) = \nu_i - (\nu_i - \nu_f)e^{-(H-z)/D} \tag{E1}$$

where $\nu_f = 0.07$ is the Poisson ratio of firn in the upper surface, $\nu_i = 0.35$ is the Poisson ratio of fully consolidated ice and $D = 32.5$ is the tuned constant. This profile approximates the observations from Schlegel et al. (2019), where we have scaled the length parameter $D$ to match our density and Young's modulus profiles as this profile was obtained at a different location (with significantly different ice-sheet and firn thickness). As it is not possible to derive a fully analytic expression for the stress profiles with this depth-dependent Poisson ratio, the longitudinal stress profiles are obtained numerically through the finite el-
ement model. Once obtained, the stresses are used to drive the propagation of the surface crevasse in the linear elastic fracture mechanics study.

We consider a dry (air-filled) crevasse, with different values of oceanwater height $h_w$ and plot the normalised crevasse penetration depth versus oceanwater height $h_w$ in Fig. E1. This figure shows that the effect of including variations in Poisson

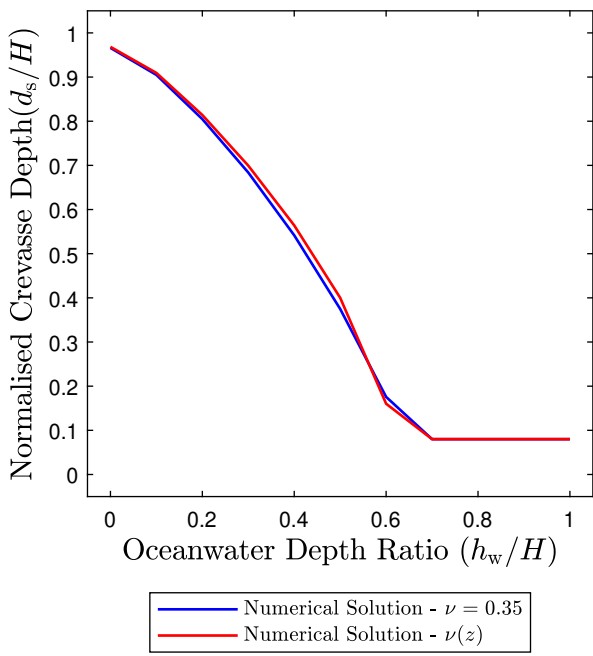

**Figure E1.** Normalised crevasse depth predictions versus oceanwater height ratio for a single isolated dry crevasse in a linear elastic ice sheet, considering homogeneous and depth-dependent Poisson Ratio.

ratio have a more limited effect compared to density and Young's modulus variations. The largest percentage difference in
crevasse depth was observed for intermediate ocean-water levels, with an increase of $6\%$ in crevasse depth with respect to
the homogeneous case when considering a depth dependent Poisson ratio, for an oceanwater height of $h_\mathrm{w} = 0.5H$. This is in
contrast to the inclusion of firn density and Young's modulus, which predict a reduction in stabilised crevasse depth for surface
crevasses in grounded glaciers. The effect of a including a depth-dependent Poisson ratio is less influential compared to density
and Young's modulus, as depth-dependent density resulted in a reduction of $20\%$ of the crevasse depth and depth-dependent
Young's modulus resulted in a reduction of $45\%$ of the crevasse depth. We therefore conclude that the inclusion of variations
in Poisson ratio does not play a significant role in crevasse propagation.

*Author contributions.* T. Clayton: Conceptualization, Methodology, Software, Validation, Formal analysis, Investigation, Data Curation, Writing - Original Draft, Visualization. R. Duddu: Conceptualization, Methodology, Writing - Review & Editing. T. Hageman: Conceptualization, Writing - Review & Editing, Supervision. E. Martínez-pañeda: Conceptualization, Resources, Writing - Review & Editing, Supervision, Project administration, Funding acquisition.

*Competing interests.* The authors declare that they have no known competing financial interests or personal relationships that could have appeared to influence the work reported in this paper.

*Acknowledgements.* T. Clayton acknowledges financial support from the Natural Environment Research Council (NERC) via Grantham Institute - Climate Change and the Environment (project reference 2446853). R. Duddu gratefully acknowledges the funding support provided by the National Science Foundation's Office of Polar Programs via CAREER grant no. PLR-1847173, NASA Cryosphere award no.80NSSC21K1003, and The Royal Society via the International Exchanges programme grant no. IES/R1/211032. T. Hageman acknowledges financial support through the research fellowship scheme of the Royal Commission for the Exhibition of 1851. E. Martínez-Pañeda acknowledges financial support from UKRI's Future Leaders Fellowship programme [grant MR/V024124/1].

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
