# Peer review of "The influence of firn-layer material properties on surface crevasse propagation in glaciers and ice shelves"

_EGUsphere, 2024_

## Author Comment (AC1)

**Authors' response to reviewers report**

**Response to Reviewer 2**

The idea that it is worth considering how the properties of firn layers could affect the stresses that control surface crevasse opening is very compelling.

We thank the reviewer for their comments, and hope to address their concerns below. For the concerns raised by using a compressible linear-elastic model, we have added extra sections to the paper showing that an incompressible viscous rheology follows similar trends, which we have attached as appendix to this peer review. Please also see the responses to peer reviewer 1 regarding these additional sections.

Comment # 1

1) However, this analysis makes the radical assumption that the stresses in an ice sheet or shelf are controlled solely by the elastic deformation of compressible ice. This is fine if one simply wants to go through a mathematical exercise, but the title, abstract and body of the paper imply that the results of this analysis applies to real ice sheets and shelves. I was particularly disturbed by the fact that the abstract does not make clear that this is an exercise based on ignoring viscous flow of ice. The fact that the Maxwell time of ice is on the order of days means that an ice sheet or shelf would have to have formed in less than a day for this analysis to be applicable.

The major conclusion of the paper is that inclusion of low-density firn produces opposite effects for idealized ice sheets versus floating ice shelves. The abstract and a cursory reading of the paper makes this seem like a general conclusion. Upon closer reading it is clear that the ice shelf result is only for a particular region close to the edge of the shelf. The authors correctly note that assumption of perfect elasticity results in compression everywhere far from the shelf edge so that no surface crevasses should result for any assumed firn densities or Young's Moduli! This is confusing because the paper only discusses analytic solutions for the stress field far from a shelf edge. To get surface crevassing on a compressible ice shelf with infinite viscosity requires bending stresses close to the edge of the shelf. The authors then use a finite element model to compute those stresses at a fixed position (250 m) from the shelf edge. At that position the predicted crevasse depth is increased by a decreasing firn density and Young's Modulus. I assume that this is a robust result but it is hard to evaluate given the information in this paper. More importantly, the paper makes it seem that this is general result based on the analytical results derived in the paper, as is clear from the opening of the "Conclusions" section:

"In this paper, we derived analytical equations for the far field longitudinal stress including the effects of surface firn layers, described by depth-varying density and Young's modulus profiles based on field data. These analytic expressions were used to perform fracture propagation studies on isolated air/water-filled surface crevasses in grounded glaciers and ice shelves ..."

This certainly gave me the wrong idea when I first read the paper.

...

It is incumbent on these authors to make a case that the assumption of perfect elasticity gives insight into the opening of surface crevasses on real ice sheets and shelves.

The reviewer raises a valid point regarding the use of linear elastic compressibility instead of a non-linear viscous rheology. The assumption of ice behaving as a linear elastic compressive material was taken, due to crevasses developing in a rapid manner below the Maxwell time-scale. The model presented is also capable of capturing this incompressible stress state by setting the Poisson ratio to $\nu \approx 0.5$. This will emulate the slow development of crevasses through ice, during which the ice has sufficient time to attain an incompressible stress state.

We note that if a Poisson's ratio of $\nu = 0.5$ is considered, the far-field stress $\sigma_{xx}$ in Eq. 1 (giving the longitudinal stresses for a compressible case without any depth-dependent material parameters) obtains

the original expression for stress derived by Weertman (1957) assuming a non-linear viscous rheology. Thus when assuming incompressiblity, the far-field stress state is independent of rheology. To address this, a new section has been added, Section 5 in the manuscript, which shows stress profiles assuming a non-linear viscous rheology and a linear elastic incompressible rheology, as well as crevasse penetration depths versus oceanwater depth ratio for a dry crevasse, considering a compressibility and incompressibility. These results show that the analytic expressions presented in our manuscript are valid the viscous stress state.

For a more in-depth discussion regarding this point, and the changes made to the manuscript to include this discussion, see the responses to Reviewer 1, comments #1 and #2.

Comment # 2

2) The other major result of the paper is that low-density firn results in smaller crevasse depths for a grounded glacier compared to a uniform ice case. The authors note that this result contradicts the previous Linear Elastic Fracture Mechanics analysis of van der Veen (1998). I suspect that the difference with the previous study is caused by the assumption of purely elastic horizontal stresses which are less extensional at the ice sheet surface than the stresses assumed by van der Veen (1998). Thus, again I am not convinced that the results of the new analysis apply to the real world.

It is correct that the inclusion of a non-linear viscous rheology results in more extensional stresses in the upper surface of the ice sheet, leading to deeper crevasses in comparison to linear elastic compressible ice. However, the inclusion of the depth-dependent density leads to reductions in both resistive stress $R_{xx}$ and lithostatic stress $\sigma_{zz}$ for both material rheologies, a point which was neglected by van der Veen (1998) who considered $R_{xx}$ to be independent of depth-varying density, which is a limitation of their work. From van der Veen (1998, page 36): "In the present model, the tensile resistive stress, $R_{xx}$, is taken constant with depth. A similar assumption is made by Rist et al. (1996) who write the full stress at depth as the sum of an arbitrary tensile surface stress and the lithostatic stress at that depth. In the notation of Eq. 4, their surface stress independent of depth corresponds to $R_{xx}$. However, if low-density firn is present, the tensile stress in this firn is probably less than that in the fully densified stronger ice at depth. In the limiting case of freshly fallen snow, it is highly unlikely that there is an appreciable surface stress. It would therefore be more appropriate to relate $R_{xx}$ somehow to the firn density assumed a proxy for the firn strength, but also accounting for the generally lower surface temperatures which may increase the strength, with the tensile stress near zero at the surface. The implication of this approach would be that surface crevasses must be initiated below the firn layer, at a depth where the ice can support a tensile stress sufficiently large to initiate fracture."

For a non-linear viscous incompressible rheology ($\nu \approx 0.5$), the resistive stress $R_{xx}$ when considering depth dependent density is given by Eq. (13), substituting $\nu = 0.5$:

$$R_{xx} = \frac{\rho_i g H}{2} - \rho_s g \frac{h_w^2}{2H} - (\rho_i - \rho_f)gD + \frac{(\rho_i - \rho_f)gD^2}{H}(1 - e^{-H/D}) \tag{1}$$

which is invariant with depth $z$ but still dependent on firn density $\rho_f$, unlike what van der Veen assumed. For example, when considering a grounded glacier of height $H = 125$m and oceanwater height of $h_w = 0.5H$, the homogeneous density case gives a value of $R_{xx} = \frac{\rho_i g H}{2} - \rho_s g \frac{h_w^2}{2H} = 404.9$ kPa, whilst the inclusion of firn density reduces the resisitive stress to $R_{xx} = 280.6$ kPa, showing that assuming this resistive stress to be independent of density depth-variations is incorrect.

For crevasses that stabilise in deeper strata, there are minimal reductions in surface crevasse depth when considering a depth-dependent density in comparison to the homogeneous case. However, as the oceanwater height increases, crevasses become shallower, and as a result, the inclusion of firn density becomes more prevalent. For example, the percentage difference in penetration depth for a dry crevasse of oceanwater height $h_w = 0.5H$ reduces to 4%, however increasing the oceanwater height to $h_w = 0.8H$

results in a percentage difference of 64% compared to the homogeneous case.

[revised manuscript text omitted]

---

## Author Comment (AC2)

**Authors' response to reviewers report**

**Response to Reviewer 1**

Clayton et al. derived analytical equations and conducted LEFM analysis to study the influence of firn-layer material properties (depth-varying density and modulus) on surface crevasse propagation in glacier and ice shelves. They found that the firn layer has a stabilizing effect on grounded glaciers (free slip boundary condition), whereas a destabilizing effect on ice shelves, with regard to fracturing and calving. The study has important implications for assessing the stability of ice sheets or ice shelves.

We thank the reviewer for their comments and positive assessment of the manuscript.

However, there are two major limitations in the assumptions of the models: i) Poisson ratio is assumed to be depth-invariant; ii) firn is assumed to be impermeable when evaluating the depth of meltwater-driven hydrofracture, neglecting the fact that meltwater will penetrate the porous firn layer instead of fracturing it. I suggest the authors reconsider the model assumption, or at least highlight the limitations, before it can receive further consideration.

Please see the responses below addressing these individual points. Paragraphs that have been added to the manuscript as a result of reviewer comments are indicated with italics in this response. Additionally, a new section and additional appendix have been added to the manuscript following reviewer feedback to showcase that the model remains appropriate for viscous/incompressible ice rheologies, and we have attached these new sections as appendix to this response.

Comment # 1

1) Why do the authors neglect depth-variations in Poisson ratio? Shouldn't Poisson ratio and Young's modulus both strongly depend on the density? Will a depth-varying Poisson ratio (which is more realistic) significantly affect the results? Below attach some references on the depth variations in Poisson ratio [1, 2, 3]. One possible way to represent the depth varying mechanical properties could be developing empirical relationships between Poisson ratio/Young's modulus, and the firn density.

[1] Schlegel, R., Diez, A., Löwe, H., Mayer, C., Lambrecht, A., Freitag, J., ... & Eisen, O. (2019). Comparison of elastic moduli from seismic diving-wave and ice-core microstructure analysis in Antarctic polar firn. Annals of Glaciology, 60(79), 220-230.

[2] Smith, J. L. (1965). The elastic constants, strength and density of Greenland snow as determined from measurements of sonic wave velocity (Vol. 167). US Army Cold Regions Research & Engineering Laboratory.

[3] King, E. C., & Jarvis, E. P. (2007). Use of shear waves to measure Poisson's ratio in polar firn. Journal of Environmental and Engineering Geophysics, 12(1), 15-21.

The reviewer raises an interesting point regarding the use of a depth invariant Poisson ratio $\nu$. In response to this, we conducted a study using the LEFM model for a dry surface crevasse subject to various ocean-water heights in a grounded glacier, considering a depth dependent Poisson's ratio and a homogeneous Poisson's ratio of $\nu = 0.35$ to determine its effect on crevasse propagation. As no analytic expressions could be derived for the depth-dependent Poisson ratio case, stress profiles used within the LEFM study were obtained through numerical finite element simulations. Results for the subsequent LEFM study are presented in Fig. 1. An exponential distribution with depth $z$ is assumed - similar to that of the density and Young's modulus distributions - with the tuned constant $D = 32.5$ and a surface Poisson's ratio $\nu_{\mathrm{f}} = 0.07$ based on the data from Schlegel et al. (2019), giving the Poisson ratio as:

$$\nu(z) = \nu_{\mathrm{i}} - (\nu_{\mathrm{i}} - \nu_{\mathrm{f}})\exp(-(H - z)/D). \tag{1}$$

[Figure]

Figure 1: Normalised crevasse depth predictions versus oceanwater height ratio for a single isolated dry crevasse in a linear elastic ice sheet, considering homogeneous and depth-dependent Poisson Ratio.

Using this depth-dependent Poisson ratio, the largest difference in crevasse depth was observed for intermediate ocean-water levels, with an increase of 6% in crevasse depth with respect to the homogeneous case for an oceanwater height of $h_{\mathrm{w}} = 0.5H$ (see Fig. 1 in this response/Figure E1 in the updated paper). This is in stark contrast with the findings for depth-dependent density and depth-dependent Young's modulus which predict a reduction in crevasse penetration depth. The effect of Poisson ratio is less influential compared to depth dependent density and Young's modulus, as depth dependent density resulted in a reduction of 20% of the crevasse depth and depth dependent Young's modulus resulted in a reduction of 45% of the crevasse depth.

The results discussing a depth-dependent Poisson ratio have been added to Appendix E. While the influence is comparatively minor, we conclude that indicating that small variations in Poisson ratio observed within ice/firn do not play a significant role, but it nevertheless an interesting finding. The full text added as appendix E reads:

*"For the crevasse propagation studies previously presented, a depth invariant Poisson ratio of $\nu = 0.35$ was assumed. However, it has been suggested that Poisson ratio also exhibits a linear dependency on ice density and therefore leads to a depth-dependent profile (Smith, 1965). Furthermore, Schlegel et al. (2019) and King and Jarvis (2007) provides a depth-dependent Poisson ratio profile based on seismic velocity measurements on ice cores. To study the effect of this depth-dependent Poisson ratio, a linear elastic fracture mechanics study is performed. We assume an exponential distribution of Poisson's ratio with depth, similar to the density and Young's modulus distributions.*

$$\nu(z) = \nu_{\mathrm{i}} - (\nu_{\mathrm{i}} - \nu_{\mathrm{f}})e^{-(H-z)/D} \tag{2}$$

*where $\nu_{\mathrm{f}} = 0.07$ is the Poisson ratio of firn in the upper surface, $\nu_{\mathrm{i}} = 0.35$ is the Poisson ratio of fully consolidated ice and $D = 32.5$ is the tuned constant. This profile approximates the observations from Schlegel et al. (2019), where we have scaled the length parameter $D$ to match our density and Young's modulus profiles as this profile was obtained at a different location (with significantly different ice-sheet*

*and firn thickness). As it is not possible to derive a fully analytic expression for the stress profiles with this depth-dependent Poisson ratio, the longitudinal stress profiles are obtained numerically through the finite element model. Once obtained, the stresses are used to drive the propagation of the surface crevasse in the linear elastic fracture mechanics study.*

*We consider a dry (air-filled) crevasse, with different values of oceanwater height $h_{\text{w}}$ and plot the normalised crevasse penetration depth versus oceanwater height $h_{\text{w}}$ in Fig. 1. This figure shows that the effect of including variations in Poisson ratio have a more limited effect compared to density and Young's modulus variations. The largest percentage difference in crevasse depth was observed for intermediate ocean-water levels, with an increase of $6\%$ in crevasse depth with respect to the homogeneous case when considering a depth dependent Poisson ratio, for an oceanwater height of $h_w = 0.5H$. This is in contrast to the inclusion of firn density and Young's modulus, which predict a reduction in stabilised crevasse depth for surface crevasses in grounded glaciers. The effect of a including a depth-dependent Poisson ratio is less influential compared to density and Young's modulus, as depth-dependent density resulted in a reduction of $20\%$ of the crevasse depth and depth-dependent Young's modulus resulted in a reduction of $45\%$ of the crevasse depth. We therefore conclude that the inclusion of variations in Poisson ratio does not play a significant role in crevasse propagation.*"

Comment # 2

2) The longitudinal stress was derived for compressible linear elasticity (Eqn.1 in the manuscript), why not viscous model? I think that a common approach, when looking at calving for example (e.g. Benn et al 2007, Ann [4]), is to calculate the background stresses from a viscous model (associated with long- term creep of the ice, and estimated perhaps from satellite-derived estimates of strain rate) instead of using an elastic model to calculate that background state. The authors might need some explanation justifying why they use linear elasticity to calculate the longitudinal stress.

[4] Benn, D. I., Cowton, T., Todd, J., & Luckman, A. (2017). Glacier calving in Greenland. Current Climate Change Reports, 3, 282-290.

The model presented is also capable of capturing this incompressible stress state by setting the Poisson ratio to $\nu \approx 0.5$. For example, for constant Young's modulus and Poisson's ratio case with $\nu \approx 0.5$, the derived analytical solution in Eq. 1 exactly matches with that of the Weertman (1957) solution [1]. As discussed in Lipovsky (2020) [2], "This initial volumetric contraction does not occur in real ice shelves because at timescales longer than the Maxwell time ice is well approximated as being incompressible." We have added a remark to the paper clarifying this:

*"The Poisson ratio $\nu$ used within our results represents ice as a linear-elastic compressible solid, which is a common assumption for rapidly propagating cracks. If the crevassing process occurs on a time-scale well below the Maxwell time-scale, ranging from hours to days depending on the strain-rate due to nonlinear viscous nature, the assumption of compressibility would be valid. Instead, if the crevassing process occurs slowly, over the span of weeks, the assumption of incompressibility would be valid; so a Poisson ratio of $\nu = 0.5$ will allow for the model derived here to be applicable over longer time-scales.*"

If we use a Poisson ratio of $\nu = 0.5$, the crevasse depths obtained for a dry crevasse subject to different values of oceanwater height $h_{\text{w}}$ are presented in Fig. 2/Figure 9. in the updated paper. This can be compared directly to the results presented in the paper for $\nu = 0.35$ (included in Fig. 2/ Figure 9. in the updated paper as dashed lines). For surface crevasses in glaciers subject to low levels of oceanwater, the penetration depth is unaffected by firn density due to crevasses stabilising in fully consolidated strata. This is the case for both linear elastic and non-linear viscous rheologies. For intermediate oceanwater heights ($0.3H < h_{\text{w}} < 0.6H$), crevasses propagate deeper when considering a non-linear viscous rheology, since stresses are more extensional. Thus, significant reductions in crevasse depths are only observed for oceanwater heights $h_{\text{w}} > 0.6H$. The largest reductions in crevasse depth are observed at $h_{\text{w}} > 0.8H$ for the non-linear viscous rheology, with a maximum percentage difference of $64\%$ between depth dependent and homogeneous results. These findings complement the results found in our paper, and follow similar

trends to the compressible stress state results. A new section has been added to the main text of the paper, including the results for crevasse propagation considering a non-linear rheology. This section provides the stress profiles resulting from our analytic expressions, compared with stress profiles from numerical finite element simulations using a visco-elastic rheology, and shows that setting $\nu = 0.5$ indeed obtains the incompressible/viscous stress state.

In addition, we also conduct linear elastic fracture mechanics studies for water-filled surface crevasses in ice shelves of height $H = 125$m and length $L = 5000$m, considering a non-linear viscous ice rheology. Similarly to the linear elastic compressible case, we consider surface crevasses at the horizontal position $x = 4750$m (250 m from the ice shelf terminus) and extract the numerical longitudinal stress profile from the finite element analysis. We plot the stabilised crevasse depth versus meltwater depth ratio for the non-linear viscous (NLV) rheology in Fig. 5 / Figure 10. of the updated paper, along with the results for linear elastic compressiblity (LE).

When comparing the stabilised crevasse depths close to the front, we note that the penetration depth is independent of ice rheology, which is in contrast to the grounded glacier. For the homogeneous density, minimal crevasse propagation is observed for meltwater depth ratios below $h_\mathrm{s}/d_\mathrm{s} < 0.6$, with full thickness propagation only occurring when fractures are close to saturation. The inclusion of the depth-dependent density results in deeper crevasse penetration depths, with minimal differences in penetration depth between the linear elastic and non-linear viscous rheology. This likely indicates that for crevasses close to the front, fracture is driven by the flotation height. For depth-dependent density, the reduction in flotation height leads to an increase in tensile stress in the upper surface, due to reductions in $R_{xx}$ and increased bending stress. In addition, the lithostatic component of longitudinal stress is reduced, leading to deeper crevasse propagation when including firn density.

We also consider the propagation of an isolated surface crevasse located in the far field region of a floating ice shelf, with results presented in Fig. 4 / Figure 11. of the updated paper. As stated previously, for the linear elastic compressible rheology the stress state is fully compressive for both the homogeneous and the depth-dependent density case, thus no crevasse propagation is observed regardless of meltwater depth ratio. By contrast, when considering the non-linear viscous rheology of ice, surface crevasses may propagate in the far field region if there is sufficient meltwater pressure present. Large increases in crevasse penetration depth are observed for meltwater depth ratios greater than $h_\mathrm{s}/d_\mathrm{s} = 0.50$, with full thickness propagation being observed close to crevasse saturation at $h_\mathrm{s}/d_\mathrm{s} = 0.95$. Similar to crevasses near the front, the inclusion of depth dependent density results in increased crevasse penetration depths compared to the homogeneous density scenario. These results have been added as Section 5 in the paper.

[1] J. Weertman, 'Deformation of Floating Ice Shelves', Journal of Glaciology, vol. 3, no. 21, pp. 38–42, Jan. 1957, https://doi.org/10.3189/S0022143000024710.

[2] B. P. Lipovsky, 'Ice shelf rift propagation: stability, three-dimensional effects, and the role of marginal weakening', The Cryosphere, vol. 14, no. 5, pp. 1673–1683, May 2020, https://doi.org/10.5194/tc-14-1673-2020.

Comment # 3

3) Once the authors start to consider meltwater within the crevasse, it confuses me that the porous nature of firn is completely ignored. LEFM no longer holds for porous material and poromechanics [5] should be considered. Could the authors at least highlight the limitations of current results (Figure 5&7 in the main text)?

[5] Coussy, O. (2004). Poromechanics. John Wiley & Sons.

The reviewer raises an intriguing point regarding the porous nature of firn with regards to meltwater pressure. In this study we have assumed that meltwater pressure is restricted solely to the fractured region. For crevasses in colder ice this is a realistic assumption. During crevasse propagation, water will seep into the firn surrounding the crevasse and start freezing, forming a thin ice layer. This thin ice layer then prevents further water from leaking into the surrounding firn. A similar effect happens near the

[Figure]

Figure 2: Normalised crevasse depth predictions versus oceanwater height ratio for a single isolated dry crevasse in a linear elastic ice sheet, considering homogeneous and depth-dependent mechanical properties for linear elastic incompressible ice ($\nu = 0.5$)

[Figure]

Figure 3: Normalised crevasse depth predictions versus meltwater depth ratio for a single isolated crevasse in a linear elastic ice sheet for an oceanwater height of $h_w = 0.8H$, considering homogeneous and depth-dependent density for linear elastic incompressible ice ($\nu \approx 0.5$)

[Figure]

Figure 4: Normalised crevasse depth predictions versus meltwater depth ratio for a single isolated surface crevasse located in the far field region ($x = 2500$m) considering a linear elastic (LE) and non-linear viscous (NLV) rheology.

[Figure]

Figure 5: Normalised crevasse depth predictions versus meltwater depth ratio for a single isolated surface crevasse located close to the ice shelf front ($x = 4750$m) considering a linear elastic (LE) and non-linear viscous (NLV) rheology.

surface, while water is able to permeate through the top layer of firn (which is typically around 0°C), ice layers (referred to as ice lenses) that prevent further water inflow form as soon as the water permeates deeper and reaches sub-zero temperatures [1,2]. We have added the following text to the paper, clarifying the implications of not including a poromechanics formulation for the firn:

*" One final limitation of our analytic models is related to water-filled crevasses. While we investigate the effects of considering depth-varying firn layer density and Young's modulus, we acknowledge that both these effects may be arising from the porosity of the firn. It is possible that water leaking from the crevasse into the surrounding porous firn. For colder ice-sheets and deeper crevasses, such that the water column is surrounded by ice of sub-zero temperatures, this assumption is reasonable, because any water that seeps into the surrounding ice/firn will freeze, creating an impermeable ice layer (i.e. ice lenses) surrounding the crevasse, which will prevent water from permeating further into the firn (Buzzard et al.,2018; Amory et al., 2024). As these ice lenses are typically very thin, these do not alter the stress state at the glacier scale. However, if temperate glaciers are considered, or conditions where water-filled crevasses do not penetrate to considerable depth, the firn/ice surrounding the crevasse might not be sufficiently cold to cause the ice lens to form. In such circumstances, the presented model will overestimate the crevasse depths obtained, as the saturated firn will reduce the effects of the water pressure within the crevasse by re-distributing this pressure over a larger region surrounding the crevasse. In future studies, we will consider the application of poro-damage phase field models (Sun et al., 2021; Clayton et al., 2022) to study fracture of saturated and unsaturated porous ice materials, and compared them with LEFM models. "*

[1] S. C. Buzzard, D. L. Feltham, and D. Flocco, 'A Mathematical Model of Melt Lake Development on an Ice Shelf', Journal of Advances in Modeling Earth Systems, vol. 10, no. 2, pp. 262–283, Feb. 2018, https://doi.org/10.1002/2017MS001155.

[2] C. Amory et al., 'Firn on ice sheets', Nat Rev Earth Environ, vol. 5, no. 2, Art. no. 2, Feb. 2024, https://doi.org/10.1038/s43017-023-00507-9.

[revised manuscript text omitted]

---

## Author Response (AR2)

**Authors' response to reviewers report**

**Response to Reviewer 1**

Comment # 1
I do not trust the basic result that firn results in shallower crevasse penetration that is asserted by the authors. They claim that van der Veen go the opposite result because he only considered the effect of low-density firn on the vertical stress ($\sigma_{zz}$) and not on the difference between the horizontal and vertical stress ($R_{xx}$). It is pretty easy to show that low-density firn has a larger effect on the magnitude of $\sigma_{zz}$ than on $R_{xx}$. To do that I considered a simple analytic treatment of the effect of low-density firn on vertical and horizontal stresses. I assume the Weertman (1957) stress state (which the authors call the incompressible state). To make the problem super simple I assume a firn layer of thickness $h_\mathrm{L}$ has a density equal to half of the density of ice in the rest of the layer which has a density $\rho_\mathrm{i}$. I start with the standard assumption that the vertical stress just equal to the weight of overburden so that below the firn layer the vertical stress magnitude is:

$$\sigma_{zz} = \rho_\mathrm{i} g (H - z) - \frac{\rho_\mathrm{i} g h_\mathrm{L}}{2} \tag{1}$$

where, as assumed in the paper under review, that H is the ice layer thickness, z is the distance above the base of the layer and g is the acceleration of gravity. It is easy to show that the difference between the vertical and horizontal stress through the layer is:

$$R_{xx} = \frac{\rho_\mathrm{i} g}{2} \left( H - h_\mathrm{L} + \frac{h_\mathrm{L}^2}{H} \right) - \frac{\rho_\mathrm{w} g}{2} \frac{h_\mathrm{w}^2}{H} \tag{2}$$

It makes sense to me that the magnitude of $\sigma_{zz}$ is reduced by $(\rho_\mathrm{i} g h_\mathrm{L})/2$ while the magnitude of $R_{xx}$ drops by only $(\rho_\mathrm{i} g)/2(h_\mathrm{L} - (h_\mathrm{L}^2)/H)$. For crevasses the open only within the firn the reduction in vertical stress magnitude is even greater. It is easy to calculate the depth of crevasse opening with the simple Nye (1955) assumption of no stress change on crevasse opening. Though that assumption is poor it allows for an analytic solution and in all cases I have seen the LEFM solution scales with the Nye solution. When you use the Nye assumption the crevasses are indeed deeper than for a case with uniform density. This little exercise makes me think there is something wrong in the solution derived in this paper though I have not taken the time to go through their analysis.

We thank the reviewer for their comment. Regarding the derived $R_{xx}$ we would like to point out a minor error (or typo) in the reviewer's derivation. Starting from the same assumption as made by the reviewer, two ice layers with different densities as shown in Fig. 1, and assuming that the vertical stress (without the seawater pressure) is equal to the cryostatic stress, we can write:

$$\sigma_{zz} = \begin{cases} -\frac{\rho_\mathrm{i} g}{2} (H - z) & \text{if} \quad H - h \leq z \leq H \\ -\rho_\mathrm{i} g (H - z) + \frac{\rho_\mathrm{i} g h}{2} & \text{if} \quad 0 \leq z \leq H - h \end{cases} \tag{3}$$

with this expression for $0 < z < H - h$ matching that of the reviewer.

The horizontal stresses are provided by:

$$\sigma_{xx} = R_{xx} + \sigma_{zz} \tag{4}$$

where, in order for the ice sheet to be in equilibrium, it is required that:

$$\int_0^H \sigma_{xx} \, \mathrm{d}z = 0 \tag{5}$$

[Figure]

Figure 1: Composition of the ice-sheet considered in the response to comment 1.

Integrating Eq. (4) over the full thickness of the icesheet, and substituting in Eq. (5) then results in the requirement that

$$\int_0^H R_{xx} \, dz + \int_0^H \sigma_{zz} \, dz = 0 \tag{6}$$

This directly relates the resistive stress $R_{xx}$ to the vertical stresses as:

$$R_{xx}H = -\int_0^H \sigma_{zz} \, dz \tag{7}$$

with $\sigma_{zz}$ given by Eq. (3). Calculating this integral of $\sigma_{zz}$, performed both by hand, and using the symbolic solver within MATLAB (see end of this peer review response), gives this integral as:

$$-\int_0^H \sigma_{zz} \, dz = \frac{1}{2}\rho_i g H(H-h) + \frac{\rho_i g}{4}\frac{h^2}{H} \tag{8}$$

Leading to a resistive stress of:

$$R_{xx} = \frac{\rho_i g}{2}\left(H - h + \frac{h^2}{2H}\right) \tag{9}$$

where we note the addition of a factor $1/2$ which was missing from the reviewer's derivation (indicated in red). If we include the seawater pressure acting on the ice sheet, this provides an offset to $R_{xx}$ as:

$$R_{xx} = \frac{\rho_i g}{2}\left(H - h + \frac{h^2}{2H}\right) - \frac{\rho_w g}{2}\frac{h_w^2}{H} \tag{10}$$

Using this expression for $R_{xx}$, the horizontal stress (responsible for crevasse propagation) is obtained from Eq. (4). Fig. 2 shows the Nye zero-stress depth, related to the depth to which crevasses are likely to propagate, with the firn layer ($h = 50$ m) and without the firn layer ($h = 0$ m) for two different seawater depths ($h_w = 0$ and $h_w = 120$ m). We have checked these stress solutions by re-calculating $R_{xx}$ based on the horizontal and vertical stress states, Fig. 3, which produces a $R_{xx}$ independent of the depth (as is correct), and have compared these results to numerical simulations obtained through COMSOL (which directly takes the density distribution, and simulates the full visco-elastic rheology of the ice until a steady stress profile is reached). Comparing these stress profiles obtained with this simple two-layer model to the results from out paper where a continuous density distribution is used shows that they capture similar trends, e.g. comparing Fig. 8 in the paper (cyan/black and pink/green lines) to Fig. 2, with both showing that using a firn layer at the top reduces the stresses in the top firn layer, while (very slightly) increasing stresses below this layer.

The above exercise emphasizes an important point that addresses the reviewer's skepticism about the correctness of our solutions. From Fig. 2, it is evident that for the no seawater case ($h_w = 0$, black line),

[Figure]

Figure 2: Horizontal stress $\sigma_{xx}$ within the ice sheet as a function of the depth coordinate $z$, obtained from the expression from Eqs. (4) and (10) and verified using the finite element software COMSOL. The total ice thickness is $H = 150$ m and for the case with the firn layer thickness $h = 50$ m and ice layer thickness $H - h = 100$ m.

the Nye zero depth is 75 m ($H/2$) without the firn layer, whereas it is 80 m with the firn layer ($H = 150$ m and $h = 50$ m). In this case, the reviewer's argument and van der Veen's original analysis is correct that the firn layer or depth varying density leads to deeper crevasse propagation. The percentage difference in crevasse depths however is small $(80 - 75)/75 \times 100 = 6.7\%$. However, if you consider the seawater case ($h_w = 120$ m, red line) in Fig. 2, the Nye zero depth is 20 m for the 'without firn layer' case and 0 m for the 'with firn layer' case. In this case, the reviewer's and van der Veen's arguments are not correct. The percentage difference in crevasse depths in this case is $(20 - 0)/20 \times 100 = 100\%$. This is the point we were trying to make in the paper, which perhaps was not so clear from the long discussions. This can be summarized with perhaps the following simple statement, which we have added to the paper - "Considering the depth-varying properties of the firn layer in grounded glaciers leads two different regimes of crevasse growth behavior. If the resistive stress $R_{xx}$ is large (e.g. a high oceanwater height), the firn layer promotes crevasse propagation and the crevasse penetrates to a greater depth than in the constant density case. In contrast, if the resistive stress is small, the firn layer has little influence on crevasse propagation. We note that if a Nye-zero stress criterion is used (e.g. for densely spaced cracks) instead of using LEFM to consider a single isolated crevasse, the firn layer hinders crevasse propagation and the crevasse penetrates to a lesser depth than in the constant density case for a low resistive stress. This can be seen in Fig. 8, where the zero-stress depth in the case of a non-linear viscous model is slightly deeper for the case including the firn density."

Comment # 2

In the response to reviewers the discussion of the new section 5 "Non-linear Viscous Incompressible Rheology" is wrong in several fundamental ways. First, the authors claim that their earlier analysis of the problem with a Poisson's ratio of 0.35 is based on a "common assumption if crevasse propagation occurs in a rapid brittle manner. . . " This completely missed to point, also made by reviewer 1, that the usual assumption is that the background stress is set by flow in the layer long before a crevasse forms. The layer is "commonly" assumed to have relaxed viscously (see Nye(1955), Weertman(1973), van der Veen

[Figure]

Figure 3: Verification of stress profiles by comparing $R_{xx} = \sigma_{xx} - \sigma_{zz}$ computed from finite element simulations in COMSOL with the analytical solution in Eq. (9).

(1998) and many others). Second, the deformation during crevasse opening is usually assumed to happen so fast that there is no viscous relaxation, but the opening magnitude certainly depends on the elastic constants (see Weertman (1980) for a clear discussing of this point) so the material cannot be treated as being incompressible. Finally, the paper still maintains that the purely elastic stress field for compressible cases are valid for fast crevasse propagation. That only works if an ice has no time to equilibrate through any viscous flow.

The reviewer's comment is more about semantics and mostly concerned with what is conventional and unconventional in glaciology literature. As we know, both linear elastic and nonlinear viscous constitutive models are phenomenological descriptions of material behavior and are always going to be approximations of the real system. We are aware and agree with the reviewer that the usual assumption is that the background stress is set by viscous flow in the layer. In fact, a proper handling of the glacier stress state must consider the theory of viscoelastic self gravitating bodies by Cathles (1975) and use the equilibrium equations based on the perturbation stress tensor (Lipovsky, 2020). However, what we have identified through finite element simulations is that the nearly-incompressible linear elastic model (i.e. Poisson's ratio $\nu \approx 0.5$) gives the same background stress field as the Maxwell viscoelastic model (with compressible linear elastic $\nu \approx 0.35$ and nonlinear viscous relations) and nonlinear viscous Stokes flow. Thus, the stress state is not sensitive to elasticity or viscosity, but rather to the assumption of incompressibility. We show this through a derivation of the longitudinal stress for elasticity in our prior paper (Sun et al., 2021). Therefore, our assumption is consistent with the assumptions of Nye, Weertman, and van der Veen with the background stress state.

Regarding the second comment from the reviewer regarding the opening magnitude depending on the elastic properties in a visco-elastic state, we agree that the short-term opening is indeed mainly a result of the elastic properties, but note that assuming ice to be incompressible does not prohibit an elastic

crevasse opening response. However, independent of material model used, none of the methods used throughout our work, or in the works from Nye, Weertman, and van der Veen have any dependence of the crack depth on the opening width. These two phenomena occur on such a different scale, with the crevasse propagating to depths of 10m-100m, whereas opening heights are typically sub-metre for newly opened crevasses. As a result, the crevasse depth calculations do not need to consider the small changes in stress due to the crevasse width as these small changes are overshadowed by the stress changes due to the presence of the crack itself. The reverse is true, however, where the crack width does depend on the depth to which the crevasse propagates, but this width is not discussed anywhere in the submitted manuscript.

Moreover, we have verified our studies with finite element simulations conducted with the phase field model (Clayton et al., 2022) in the current paper, and also with the cohesive zone model in another recently published paper (Gao et al., 2023). The benefit of these computational models is that they do not make any explicit assumptions and consider elastic and viscous processes to occur as dictated by the Maxwell time scales, which will be spatially varying due to the dependence of viscosity on strain rate. Please see our recent work on the role of viscous deformation on turbulent hydrofracture in glaciers (Hageman et al., 2024). In response to the reviewer's concerns we have made some minor text changes in Section 5, that should clarify our methodology to mathematical glaciologists that use the approaches of Nye, Weertman, and van der Veen.

Cathles, L. M. (1975). Viscosity of the Earth's Mantle. Princeton University Press. http://www.jstor.org/stable/j.ctt13x0t47)

Lipovsky, B. P.: Ice shelf rift propagation: stability, three-dimensional effects, and the role of marginal weakening, The Cryosphere, 14, 1673–1683, https://doi.org/10.5194/tc-14-1673-2020, 2020.

Clayton, T., Duddu, R., Siegert, M. & Martínez-Pañeda, E.: A stress-based poro-damage phase field model for hydrofracturing of creeping glaciers and ice shelves. Engineering Fracture Mechanics 272, 108693 (2022).

Gao, Y., Ghosh, G., Jiménez, S. & Duddu, R.: A Finite-Element-Based Cohesive Zone Model of Water-Filled Surface Crevasse Propagation in Floating Ice Tongues. Computing in Science & Engineering 25, 8–16 (2023).

Hageman, T., Mejía, J., Duddu, R. & Martínez-Pañeda, E.: Ice viscosity governs hydraulic fracture that causes rapid drainage of supraglacial lakes. The Cryosphere 18, 3991–4009 (2024).

**Response to Reviewer 2**

We thank the reviewer for their recommendation to accept the paper as-is.

**Derivation of assumed model from reviewer 1**

Declaring symbols, add assumptions on H and h

```
syms h H rho_i z g
assume(H>0)
assume(H>h>0)
```

Defining the vertical stresses based on the hydrostatic stress state, with expressions used for the bottom and top parts of the icesheet

```
s_zz = piecewise(0<=z<=H-h, -rho_i*g*(H-z) + rho_i*g*h/2, ...
                 H-h<z<H, -rho_i*g/2*(H-z)   )
```

s_zz =

$$
\begin{cases}
\dfrac{g\,h\,\rho_i}{2} - g\,\rho_i\,(H-z) & \text{if } 0 \le z \wedge h + z \le H \\[2mm]
-\dfrac{g\,\rho_i\,(H-z)}{2} & \text{if } z < H \wedge H < h + z
\end{cases}
$$

```
s_zz_nofirn = -rho_i*g*(H-z)
```

s_zz_nofirn = $-g\,\rho_i\,(H-z)$

Integrate these stresses over the full icesheet thickness

```
integral = int(s_zz, 0, H)
```

integral =

$$
-\frac{g\,h^2\,\rho_i}{4} - \frac{H\,g\,\rho_i\,(H-h)}{2}
$$

```
integral_nofirn = int(s_zz_nofirn, 0, H)
```

integral_nofirn =

$$
-\frac{H^2\,g\,\rho_i}{2}
$$

and divide by -1/H to obtain the value of R_xx

```
R_xx = -expand(1/H * integral)
```

R_xx =

$$
\frac{H\,g\,\rho_i}{2} - \frac{g\,h\,\rho_i}{2} + \frac{g\,h^2\,\rho_i}{4\,H}
$$

```
R_xx_nofirn = -expand(1/H * integral_nofirn)
```

R_xx_nofirn =

$$
\frac{H\,g\,\rho_i}{2}
$$

Substitute back to obtain s_xx

```
s_xx = expand(s_zz + R_xx)
```

s_xx =

$$
\begin{cases}
g\,\rho_i\,z - \dfrac{H\,g\,\rho_i}{2} + \dfrac{g\,h^2\,\rho_i}{4\,H} & \text{if } 0 \le z \wedge h + z \le H \\[2ex]
\dfrac{g\,\rho_i\,z}{2} - \dfrac{g\,h\,\rho_i}{2} + \dfrac{g\,h^2\,\rho_i}{4\,H} & \text{if } z < H \wedge H < h + z
\end{cases}
$$

```
s_xx_nofirn = expand(s_zz_nofirn + R_xx_nofirn)
```

s_xx_nofirn =

$$
g\,\rho_i\,z - \dfrac{H\,g\,\rho_i}{2}
$$

Plot the obtained relations for different firn thicknesses,

```
figure
tiledlayout('flow')
for h_num = [10, 20, 50, 100]
    s_xx_num = subs(s_xx, [h, H, rho_i, g],[h_num, 150, 910, 9.81]);
    s_zz_num = subs(s_zz, [h, H, rho_i, g],[h_num, 150, 910, 9.81]);
    s_xx_nofirn = subs(s_xx_nofirn, [h, H, rho_i, g],[50, 150, 910, 9.81]);
    s_zz_nofirn = subs(s_zz_nofirn, [h, H, rho_i, g],[50, 150, 910, 9.81]);

    nexttile
    fplot(s_xx_num*1e-6, [0, 150],'k')
    hold on; view([90 -90]); xlabel('z [m]'); ylabel('\sigma [MPa]')
    fplot(s_zz_num*1e-6, [0, 150],'r')
    fplot(s_xx_nofirn*1e-6, [0, 150],'k-.')
    fplot(s_zz_nofirn*1e-6, [0, 150],'r-.')
    title("h="+string(h_num))
end
leg = legend('\sigma_{xx}','\sigma_{zz}','NumColumns',2);
leg.Layout.Tile = 'south';
```